# Ex vivo validation of magnetically actuated intravascular untethered robots in a clinical setting
Leendert-Jan W. Ligtenberg [1], Nicole C. A. Rabou[1], Constantinos Goulas[2,3], Wytze C. Duinmeijer [1,3], Frank R. Halfwerk [1,3], Jutta Arens[1,3], Roger Lomme[4], Veronika Magdanz [5], Anke Klingner [6], Emily A. M. Klein Rot[7], Colin H. E. Nijland[7], Dorothee Wasserberg[7,8], H. Remco Liefers[3], Pascal Jonkheijm[7,8], Arturo Susarrey-Arce[9], Michiel Warlé[10] & Islam S. M. Khalil [1,3] ✉

Intravascular surgical instruments require precise navigation within narrow vessels, necessitating maximum flexibility, minimal diameter, and high degrees of freedom. Existing tools often lack control during insertion due to undesirable bending, limiting vessel accessibility and risking tissue damage. Next-generation instruments aim to develop hemocompatible untethered devices controlled by external magnetic forces. Achieving this goal remains complex due to testing and implementation challenges in clinical environments. Here we assess the operational effectiveness of hemocompatible untethered magnetic robots using an *ex vivo* porcine aorta model. The results demonstrate a linear decrease in the swimming speed of untethered magnetic robots as arterial blood flow increases, with the capability to navigate against a maximum arterial flow rate of 67 mL/min. The untethered magnetic robots effectively demonstrate locomotion in a difficult-to-access target site, navigating through the abdominal aorta and reaching the distal end of the renal artery.

Untethered magnetic robots (UMRs) have the potential to navigate through bodily fluids for surgical or therapeutic procedures, such as targeted therapy and material removal. When operating in vitro, the navigation of UMRs is often greatly simplified by a controlled environment in which a detailed analyses of one or more important physical effects is studied inside Petri dishes or test tubes. This type of experiment has allowed us to advance our knowledge about the incorporation of a specific physical intelligence into UMRs, which is significantly important at small scales, allowing them to be used as end-effectors of wireless manipulation systems[1,2]. Microactuation[3], high-precision transportation and cargo delivery[4,5], gamete transport[6,7], microassembly[8,9], diagnosis[10], material removal, and targeted drug delivery[11] have been demonstrated in vitro at a number of scales[12]. While these promising experiments have indeed demonstrated the potential of UMRs across various technologies and therapies, it is important to acknowledge that they currently fall short of replicating the intricate conditions found within living organisms. As a result, the full extent of their capabilities and limitations remains unexplored.

It is unlikely that UMRs can effectively be used in vivo unless multiple hurdles are addressed simultaneously, such as wireless power[13], locomotion[14,15], localization[16,17], control robustness[18], and biocompatbility[19]. Consider, for example, a scenario where reaching a particular location proves challenging through conventional tethered methods (Fig. 1A)[20]. In this case, the UMR would be inserted in either a fluid-filled lumen or soft tissue, allowing access to the whole human body by swimming through bodily fluids drilling through tissue, or both. This is most practically done through a UMR designed with a chiral geometry (e.g., screw-shaped or helical body), which can be driven by homogeneous rotating magnetic fields[21–24]. To achieve the objective of reaching its location, the UMR must effectively harness sufficient mechanical energy for its locomotion while contending with the dynamics of blood circulation. To reach the desired location in Fig. 1A, the UMR must be

[1]Department of Biomechanical Engineering, University of Twente, 7500 AE Enschede, The Netherlands. [2]Department of Design Production and Management, University of Twente, 7500 AE Enschede, The Netherlands. [3]Technical Medical Centre, University of Twente, 7500 AE Enschede, The Netherlands. [4]Radboud University Medical Center, 6525 GA Nijmegen, The Netherlands. [5]Department of System Design Engineering, University of Waterloo, ON N2L 3G1 Waterloo, Canada. [6]Department of Physics, German University in Cairo, New Cairo 11835, Egypt. [7]LipoCoat B.V., 7521 AG Enschede, The Netherlands. [8]Laboratory of Biointerface Chemistry, TechMed Centre, University of Twente, 7500 AE Enschede, The Netherlands. [9]Mesoscale Chemical Systems, MESA+ Institute, University of Twente, 7500 AE Enschede, The Netherlands. [10]Department of Surgery, Division of Vascular and Transplant Surgery, Radboud University Medical Center Nijmegen, Nijmegen, The Netherlands. ✉e-mail: i.s.m.khalil@utwente.nl

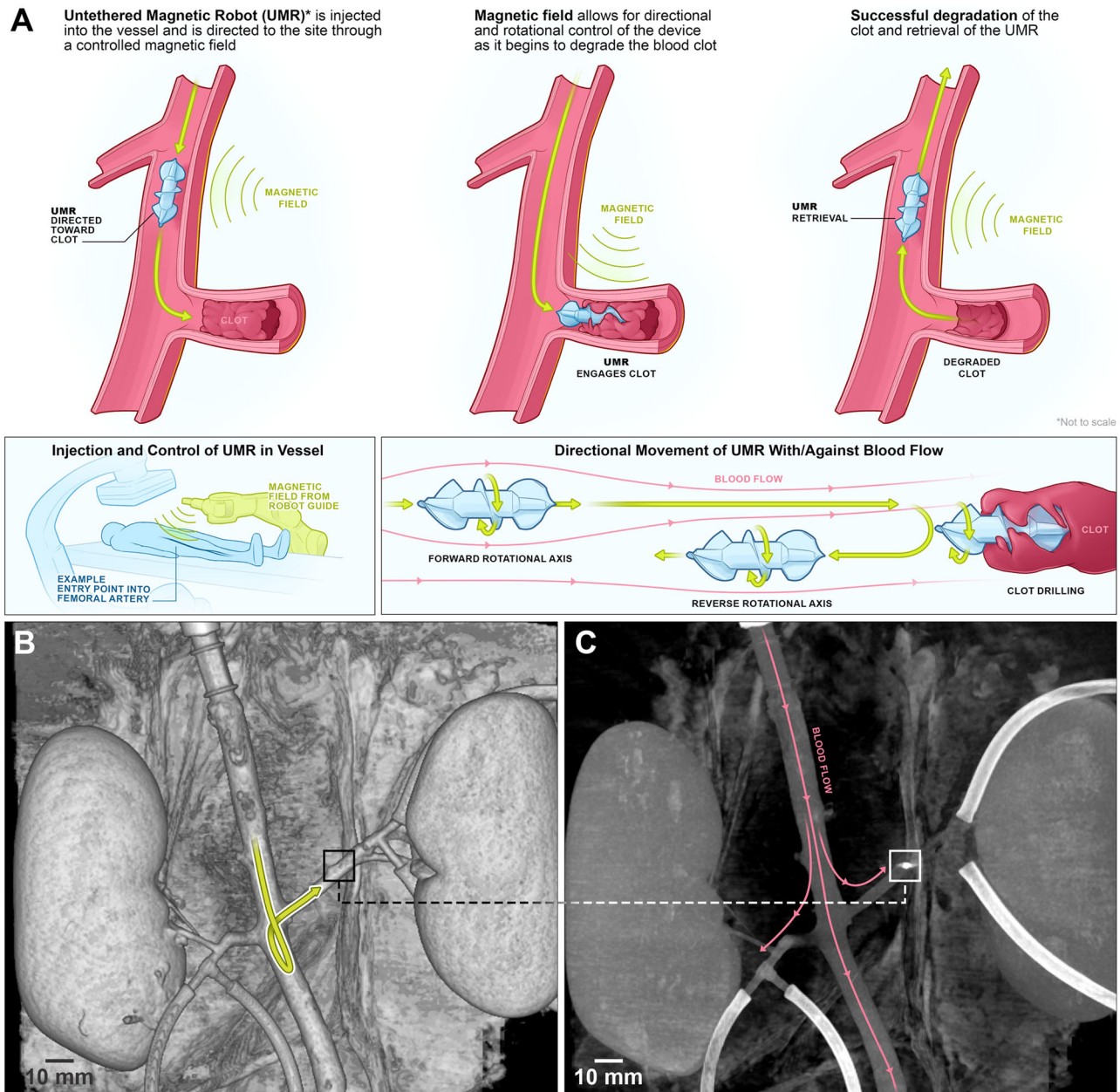

**Fig. 1 | The untethered magnetic robot (UMR) can swim inside the natural pathways of a porcine aorta model under controlled conditions, enabling interventions and retrieval with minimal incisions. A** The wireless actuation and non-invasive localization of UMRs are achieved through a robotic platform, consisting out of an external rotating magnetic field and a C-arm imaging system. UMRs navigate both with and against the blood flow for various interventions. **B, C** A 9-mm-long UMR moves both with and against the blood flow inside the abdominal aorta and is then guided to swim within the left renal artery. The UMR's location is highlighted by squares in the cone-beam computed tomography scans, and its trajectory is depicted by the yellow arrow (Movie S1).

steered controllably at bifurcations and driven with the optimal rate at which maximum propulsive thrust is achieved. This scheme requires that we localize the UMR, reconstruct its physical surroundings, and achieve an acceptable level of biocompatibility and control robustness using a robotic platform at a scale relevant to clinical use. Even then, the true capabilities of these devices can only be conclusively demonstrated through ex vivo and in vivo experimental results. This is due to the complex interplay of multiple physical effects (such as the wall effect, blood flow, vessel bifurcations, and magnetohydrodynamic coupling), which cannot be accurately replicated in vitro.

To fully harness the anticipated potential of UMRs in biomedicine, it is essential to address the challenges mentioned earlier and work toward developing clinically feasible techniques. In recent years, there has been some progress in realizing wireless control of UMRs in vivo. Niedert et al. have successfully maneuvered a tumbling microrobot within a live mouse's colon, employing an external periodic magnetic field and an ultrasound imaging system. However, a notable challenge lies in the restricted field-of-view during localization, consequently confining the procedure's applicability to smaller animals exclusively[25]. In contrast, Vonthrom et al. have utilized a clinical magnetic resonance imaging (MRI) system (large enough to provide a strong field gradient over the human body) to localize, steer, and control microrobots and microdevices[26]. In this case, MRI systems generate wide field-of-view images for localization and control with adequate resolution. However, the locomotion of torque-driven UMRs is limited by the MRI systems, which allow for partial control as field gradients can only be scaled. Similarly, Tiryaki et al.[27] have shown an MRI-powered magnetic miniature capsule robot capitalizing on acoustic streaming forces generated by high-intensity focus ultrasound for controlled drug release. However, for ferromagnetic torque-driven UMRs actuated by homogeneous magnetic

fields, MRI systems become unfeasible as a high magnetic field associated with imaging would interfere with the magnetization and magnetic response. Therefore, there is a need to mitigate the influence of the field associated with the imaging instrumentation on the wireless actuation. Bakenecker et al. have introduced a magnetic robot designed for precise navigation through a human cerebral aneurysm phantom using a magnetic particle imaging (MPI) scanner, enabling untethered aneurysm coiling[24]. This presents a novel approach to intervention without the need for iodine-based contrast agents or ionizing radiation. The robot, coated with magnetic particles, facilitates magnetic actuation but lacks environmental visualization capabilities with MPI. This is because MPI is specifically designed to detect the three-dimensional distribution of superparamagnetic iron-oxide nanoparticles, rather than the surrounding environment. Servant et al. have used feedback from an optical fluorescence imaging system to control a swarm of functionalized artificial bacterial flagella (ABFs) in vivo[28]. In this scenario, the ABFs are functionalized with near-infrared fluorophores for the purpose of tracking within the intraperitoneal cavity of a small anesthetized animal. The animal is enclosed by coils to facilitate wireless actuation. While this optical technique facilitates comprehensive whole-body imaging, the challenge remains in effectively tracking UMRs within deep tissues and vessels, posing a notable hurdle.

Here we translate UMRs into ex vivo trials and achieve directional control inside a porcine aorta model with varying blood vessel diameter. Figure 1B, C show the planned path of the UMR and the arterial blood flow, respectively (Movie S1 displays the trajectory followed by the UMR). This planned trajectory ensures that the UMR navigates both against and with the arterial flow, progressing toward the distal end of the renal artery, with validation performed using cone-beam computed tomography (CBCT) scans. To achieve this level of control, we assess the UMR's swimming behavior in a porcine aorta model in a clinical setting (Fig. 2A, B) by using wireless actuation and localization systems that can be effectively scaled up (Fig. 2C–F). Initially, we undertake the design and characterization of UMRs that possess the ability to navigate through the whole porcine aorta model. The magnetic behavior of these UMRs in response to an external actuating magnetic field is assessed through an in vitro blood vessel model. Notably, only extended-duration characterization experiments are conducted in vitro. The predictions for optimal actuation inputs—specifically, actuation frequency and magnetic field strength—are made using ultrasound images and are correlated with the fluid properties. Subsequently, with these calculated inputs, we successfully showcase that UMRs are capable of controlled movement within confined spaces. This controlled movement enables the UMRs to navigate the interior of blood vessels in the porcine aorta model while minimizing contact with the vessel walls (Fig. 2G, H). In order to assure hemocompatibility of the UMRs a lipid-based coating was applied and various assay were carried out to verify this hemocompatibility. Figure 2I shows micrographs of coated UMR material after incubation with fluorescently labeled fibrinogen. Fibrinogen adsorption on the coated samples is reduced by 95% compared to the uncoated samples. Figure 2J shows micrograph images of biofilms formed by *Staphylococcus aureus* on coated and uncoated samples, with a reduction of more than 99%. Figure 2K, L compare the fibrin generation in time, of platelet poor plasma in contact with coated versus uncoated samples. Fibrin generation was delayed by > 6 min for the coated samples compared to the uncoated samples. Together with further assays (see Methods) the data indicates that no detrimental effect of coated UMRs is expected during in vivo applications.

## Results
### Ex vivo model and robotic platform
Leveraging the detailed understanding of the aorta's anatomy and physiology (see Methods), we are able to evaluate the swimming capabilities of the UMRs. By conducting straight runs within the abdominal aorta, we can observe the UMRs' swimming behavior both against and with the blood flow. Additionally, the setting of the renal aortic side branch offers a suitable environment for assessing the UMRs' ability to achieve directional control (Fig. 1B, C). The proximal and distal ends of the abdominal aorta are connected to a peristaltic pump for blood circulation at controlled pulsatile flow rate in the $15 \leq \dot{Q} \leq 260$ mL/min range. The average flow rate inside the abdominal aorta is 2.9 L/min[29], presenting a challenge for the UMR. However, enhancing the UMR's propulsive thrust is feasible by increasing its magnetic moment and strengthening the external magnetic field. Constructed with a screw-shaped body made through additive techniques, the UMR incorporates a fixed permanent magnet. Augmenting the proportion of ferromagnetic material is anticipated to refine its flow response (Fig. S1), enabling greater magnetic torque under a given magnetic field, while concurrently reducing buoyancy. Although the physiological flow rate surpasses the range simulated in our ex vivo model, a UMR deployed for targeted blood flow restoration due to blockage is unlikely to encounter such high flow rates in practice.

Through the addition of a lipid-based coating, the UMR can attain self-sufficiency and maintain cellular viability[30]. The lipid-based coating has been shown to be highly hemocompatible, activating neither the clotting pathways nor the complement system (see Methods and Fig. S2). Because of the affixed miniature permanent magnet, we avoid the need to create strong magnetizing fields. The UMR is inserted into the model from the proximal end of the abdominal aorta through a large bore cannula (Fig. 2A). An intra-aortic 3D-printed filter is connected to the distal end of the aorta (Fig. 2B) to retain the UMR when the applied blood flow is much greater than its propulsive thrust, or when the UMR is not magnetically coupled with the rotating permanent magnet (RPM) actuator, temporarily. The ex vivo model and the RPM actuator are placed between an X-ray source and a detector, as shown in Fig. 2C–F. As the X-ray beam traverses through the model, the affixed UMR magnet, and the RPM actuator, its intensity is diminished. This attenuation occurs as the X-ray travels from the source to the detector array. The acquired CBCT-scan data enables us to reconstruct the internal structure of the model in the dimensions with precision, as shown in Fig. 1B and C, which would be useful in examining the positioning accuracy after actuation.

In contrast, X-ray Fluoroscopy images are gathered online with good resolution and at adequate frame rate (5 frames per second) for direct teleoperation, allowing the UMR to swim controllably under the influence of external inputs given directly by a clinician. Figure 3A shows an X-ray Fluoroscopy image during a straight run inside the abdominal aorta. A rotating magnetic field gradient is generated by the RPM actuator, which is directly teleoperated based on the gathered X-ray Fluoroscopy images. Our robotic platform (i.e., C-arm and wireless manipulation system) is configured such that the UMR, its physical surroundings, and the RPM actuator are captured in each X-ray Fluoroscopy image, as shown in Fig. 3A. This is accomplished by using an oblique angle for the X-ray source, and the detector array (Fig. 3B). The source and the detector of the C-arm imaging system are kept at an oblique angle of 20° with the $\mathbf{z}$ axis (in the frame of reference in Fig. 3). This setup enables the captured X-ray Fluoroscopy images to clearly display both the UMR and the RPM actuator, thereby enhancing the intuitiveness of teleoperation. Furthermore, the oblique orientation of the C-arm offers the RPM actuator an expanded workspace, minimizing the potential for interference with the detector array. Figure 3C illustrates the UMR's geometry, achieved by introducing a radiocontrast agent into an in vitro model (refer to Methods), enhancing the visibility of the radiolucent structure.

Figure 3A shows the configuration of the RPM actuator and the position of the UMR during a straight run against arterial flow. In this case, the RPM rotation axis, $\mathbf{\Omega}_{act}$, is oriented parallel to the centerline of the abdominal aorta, and its translational velocity is controlled such that it remains in sync with the UMR. Under clinically relevant radiation doses, the low contrast resolution allows for a detectable signal from the attached radiopaque magnet. Consequently, only the magnet of the UMR becomes visible in the X-ray Fluoroscopy images in Fig. 3A, D, and E. Although controlling the UMR is challenging without orientation information at this radiation level (Fluoroscopy dose rate of: 0.35 mGy cm$^2$ s$^{-1}$), the magnetic torque would ultimately allow the UMR to align with the RPM rotation axis. This is the method used in Fig. 3D to steer the UMR and enter the left renal

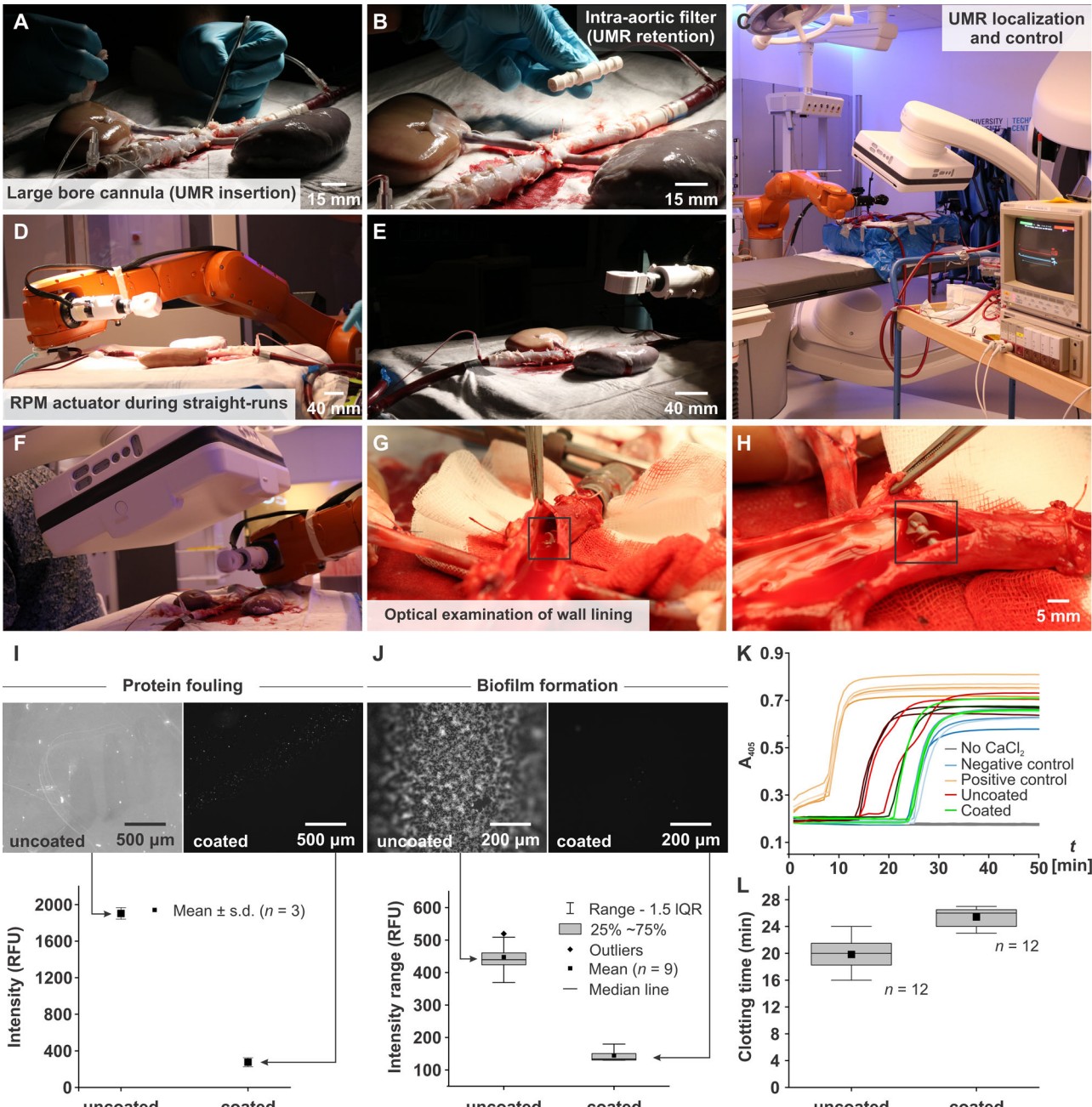

**Fig. 2 | Ex vivo trials are conducted using a porcine aorta model. A** The ex vivo organs are harvested and connected to a circulation pump prior to each motion control session. **B** Placement of an intra-aortic 3D-printed filter allows the untethered magnetic robot (UMR) to remain inside the aorta irrespective of the ongoing blood circulation. **C–F** Wireless magnetic actuation and C-arm imaging systems enable control and localization, respectively. **G, H** Optical examination of the internal wall lining of the vessels shows no risk of damage by the UMR. The black squares indicate the UMR. **I–L** Hemocompatibility tests included **I** protein fouling, **J** biofilm formation, and **K, L** coagulation evaluation.

artery, as shown in Fig. 3E. Additionally, it is worth noting that the orientation of the UMR can be determined by incorporating radiopaque markers into it or by utilizing a permanent magnet with a nonuniform aspect ratio, enabling orientation detection.

## Hemobiocompatability of UMRs

Previously, we have shown that cell adhesion, cell morphology, focal adhesion formation, cell proliferation, and cell differentiation potential remain unaffected by the coating components[31]. Here, we investigate biocompatibility in vitro in terms of protein fouling, biofilm formation, and various hemocompatibility assays. These tests are conducted using coated UMR- and other materials. It is well-established that the initial step of the surface-activated (intrinsic) pathway of the clotting cascade involves the interaction of a protein (factor XII) with a foreign substrate. Similarly, the complement system is activated through protein-substrate interactions[32]. Hence, the affinity of proteins for a material is believed to be a determinant of a material's hemocompatibility[33]. To evaluate this critical protein-material affinity, fibrinogen (clotting Factor I) is selected for use in a protein fouling assay. Microscopy of coated UMR material after incubation with fluorescently labeled fibrinogen revealed a 95% reduction in protein adsorption compared to uncoated UMR material (Fig. 2I). This reduction in protein fouling is expected to diminish the activation of the clotting cascade and complement system during in vivo use.

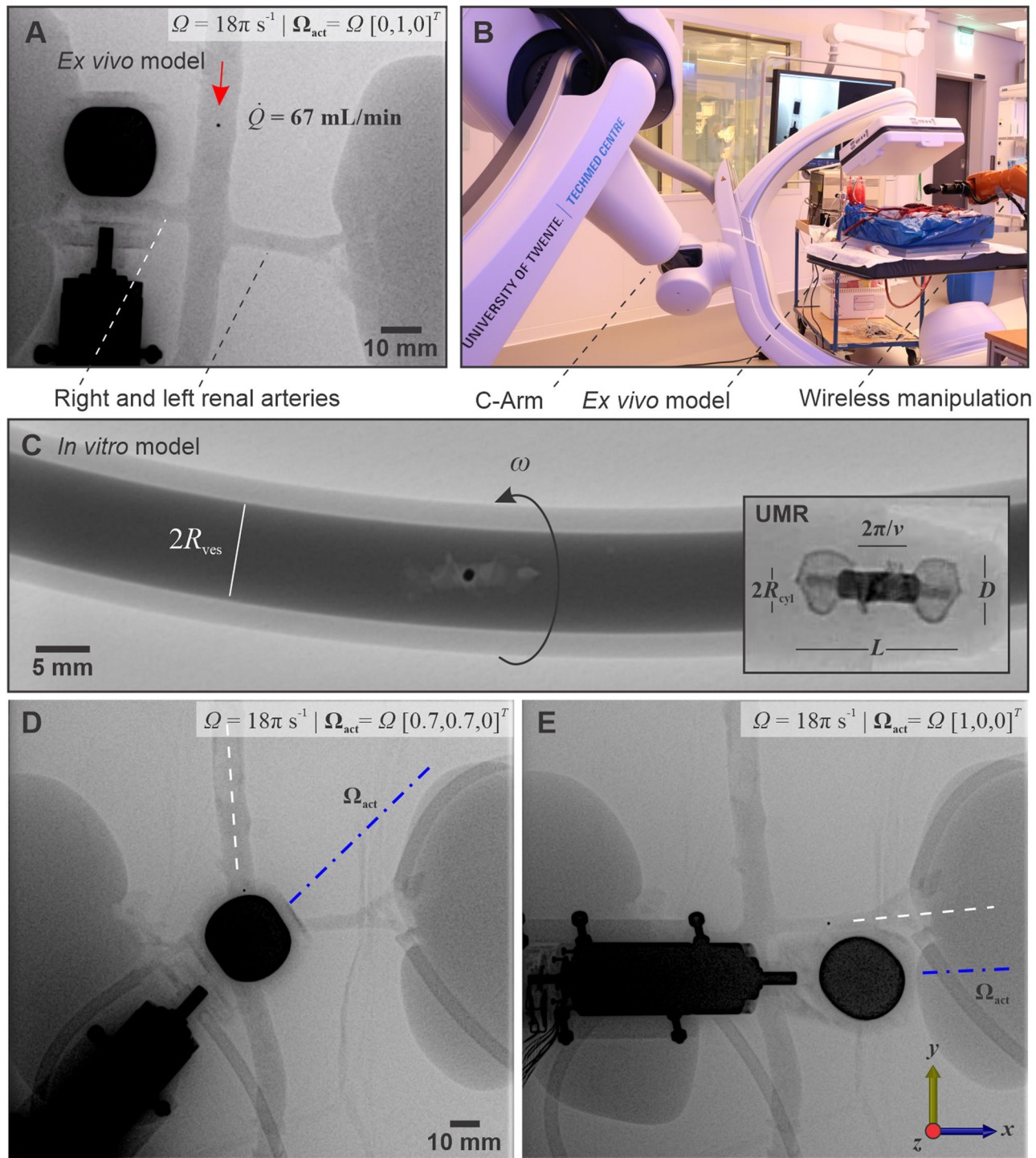

**Fig. 3 | UMRs are employed to navigate vascular pathways for interventions.**
**A** X-ray fluoroscopy images are gathered online to detect the UMR, the rotating permanent magnet (RPM) actuator, and the physical surroundings using clinically relevant radiation settings. **B** Our robotic platform consists of a C-arm imaging and permanent magnet robotic systems. **C** To show the shape of the UMR, a radio-contrast agent is injected into an in vitro model. **D**, **E** The process of steering and maneuvering the UMR within the left renal artery is achieved by manipulating the rotation axis of the RPM about the **z** axis.

Bacterial infections pose a prevalent risk when using medical devices[34]. Bacteria that adhere to medical devices often lead to severe complications and can contribute to the formation of biofilms, which are challenging to treat and can promote antibiotic resistance[35]. Therefore, we conducted an assessment to inhibit bacterial attachment and subsequent biofilm formation on coated samples, initially focusing on attachment Fig. 2J. The results indicated that the coated samples exhibited a reduction of over 99% in attached bacteria compared to the uncoated samples. This underscores the

potential of using coated UMRs to reduce the risk of infection when compared to their uncoated counterparts. In the case of short dwell time UMRs, the primary infection risk appears to stem from bacterial introduction during insertion rather than supporting biofilm formation.

Another common hemocompatibility assay is the fibrin generation test[36,37]. It is routinely used to assess the formation of fibrin fibres, which, together with platelets, constitute the final stage of the clotting cascade, the blood clot. For the independent assessment of the intrinsic pathway, fibrin

formation was tested using platelet poor plasma (PPP) isolated from freshly drawn, citrated whole blood. Coated and uncoated UMR material was immersed in citrated PPP and, subsequently, the clotting cascade was re-initiated by reconstituting $Ca^{2+}$ to physiological levels (Fig. 2K, L). Comparing coated with uncoated UMR material, a maximum of 6 min delay in fibrin formation was observed for the coated material. Thus, the coated material clearly delays the clotting via the intrinsic pathway, and, therefore, coated UMRs are expected to carry a reduced embolic risk in vivo compared to uncoated UMRs.

LipoCoat 4AC coated PU catheter material was subjected to a variety of hemocompatibility tests carried out by a company dedicated to hemocompatibility testing of blood-contacting devices, HAEMOSCAN BV. Hematology tests were carried out, assessing platelet and red blood cell counts, quantification of hemoglobin, material-induced hemolysis, coagulation, platelet activation, as well as inflammation and complement system. The coated samples passed all hemocompatibility tests at levels comparable, if not better, than the negative control. The entirety of the data discussed above indicates coated samples to be highly hemocompatible and to expect no issues with regard to coagulation or complement activation during in vivo use.

## Wireless locomotion of UMRs in arterial flow

The movement of screw-shaped UMRs within low-$Re$ (in the range between 0.5 and 20; refer to Methods) blood flow depends on three physical phenomena. The initial effect involves drag anisotropy, crucial for generating a net propulsive thrust through lateral movement or rotation around the long axis of the UMR. This characteristic is attained through the screw-shaped configuration of the body (refer to Methods). In this case, the helical pitch, $2\pi/\nu$, is a crucial design feature that can be optimized to yield sufficient thrust. The second effect is the impact of the lumen wall. This wall effect becomes relevant within restricted spaces and is anticipated to increase the swimming speed for a given helical pitch of the UMR. However, within confinement, the swimming speed demonstrates an increase up to a local maximum and subsequently declines to a local minimum as the helical pitch decreases[38]. As a result, when encountering blood vessel diameters of different sizes, the interplay between drag anisotropy and confinement effects would lead to fluctuations in the UMR's swimming speed as it advances. The third physical phenomenon is the magnetic interaction between the RPM actuator and the UMR. When UMRs are directed to navigate toward a specific vessel, such as selecting one of three pathways at a bifurcation in the model, it involves controlling the plane of rotation of the magnetic field through the rotation axis of the RPM. A periodic torque about the long axis of the UMR drives the time-averaged propulsion, while another torque in-plane (within the $xy$ plane about the $z$ axis) is responsible for steering. Note that the ex vivo model is naturally constrained to lie on the horizontal plane (Fig. 4A), and therefore only these torques are sufficient to navigate and target any location of interest. Taking these effects into consideration results in a managed reaction of the UMR within the vessels. However, accessing the renal artery could remain challenging if the alignment between the UMR's long axis and the local tangent at the entry point of the centerline is not precise. In such cases, only UMRs with dimensions slightly smaller than those of the abdominal aorta and renal artery might have the potential to reach specific points of interest.

The distance between the UMR's long axis and the blood vessel's centerline varies with the UMR-RPM gap. Adjusting the UMR-RPM gap is achievable by moving the RPM closer to the vessel, which in turn influences the UMR's proximity to the vessel's centerline. Figure 4B illustrates the influence of this gap on a UMR inside porcine blood, visualized using ultrasound images. The average in vivo gap from the abdominal aorta to the skin is ~10 cm to 12 cm. Therefore, this UMR-RPM gap is limited to ~0.1 m, generating a sufficient magnetic field to propel the UMR. Alternatively, further reducing the UMR-RPM gap can enhance the applied field in other parts of the body where blood vessels are more superficial. This control input becomes particularly valuable when managing a UMR at the onset of step-out (i.e., the frequency beyond which the UMR cannot keep pace with the

actuating field). By moving the RPM actuator closer to the vessel, the step-out frequency in such cases can be elevated. The step-out frequency of two UMRs (9-mm- and 12-mm-long) is shown in Fig. 4C, for an RPM-UMR gap of 0.1 m. Associated with the increase in the actuation frequency of the RPM is a linear increase in the swimming speed of the UMR in blood, $U$, and a similar response is observed in water $U_N$. Slightly below step-out (i.e., 10 Hz), the swimming speed of the 12-mm-long UMR is greater than that of any other actuation frequency, making it favorable for actuation. In contrast, the 9-mm-long UMR boasts a wider frequency range and can be actuated at frequencies of up to 28 Hz. While reducing the gap between the UMRs and the RPM actuator may not practically enhance the step-out frequency due to physical constraints in this body region, increasing the magnetic moment using a stronger magnet is a feasible solution.

In contrast to swimming in water, the interaction between blood and UMR is not solely elastic, as depicted in Fig. 4D. In the case of blood with a specific ratio of serum viscosity to total viscosity denoted as $\beta$, the UMR's swimming speed diminishes with higher fluid relaxation, indicated by $De$. The application of the Oldroyd-B model (outlined in the Methods section) offers predictions for the UMR's swimming speed across a spectrum of blood solvent viscosity ratios and relaxation values. If the viscosity of the blood were to increase to the point where $\beta$ approaches zero, the resultant swimming speed would also tend toward zero. In a potential medical intervention scenario (as shown in Fig. 1A), a UMR might need to be moved toward a blood clot to reinstate local flow. In this instance, it is more instructive to predict its response using our model. Our UMR speeds, scaled by their speed in a Newtonian fluid ($U/U_N$), are noticeably slower in clots compared to their speeds in blood. Therefore, for our experiments, it suffices to demonstrate controlled locomotion toward a specific location of interest, and potentially facilitate the release of a drug to reinstate the flow.

Figure 4E presents the projected UMR swimming speed as a function of the normalized wavenumber $\nu$ and cylinder-to-vessel ratio $R_{cyl}/R_{ves}$. Smaller normalized wavenumber values correspond to increased speed. Both the 9-mm-long UMR and the 12-mm-long UMR exhibit an average normalized wavenumber of ~2.2. In narrower vessels, there is an observed speed increase compared to wider vessels. Consequently, higher speeds are anticipated in the renal artery compared to the aorta. In the aorta, the small and large UMRs possess cylinder-to-vessel ratios of 0.32 and 0.42, resulting in predicted speeds of 6 mm/s and 12 mm/s, respectively. Conversely, in the renal artery, the small and large UMR feature cylinder-to-vessel ratios of 0.56 and 0.75, leading to predicated speeds of 18 mm/s and 74 mm/s, respectively.

We compare the observed swimming speeds when the UMRs are allowed to move both against and with the blood flow. Straight runs of the UMR along the abdominal aorta of the ex vivo model are conducted at actuation frequencies below the step-out threshold, aiming to achieve maximum propulsive thrust. Figure 5 illustrates the trajectory taken by the same 12-mm-long UMR during a straight run at an actuation frequency of 9 Hz. In this trial, the run begins by propelling the UMR against the flow, originating from the distal end of the abdominal aorta and progressing toward its proximal end. As the UMR moves along its path, it encounters varying flow velocities. Notably, the blood flow speed past the bifurcation of the renal arteries exceeds that of any other parts of the ex vivo model. Consequently, at $t = 5$ s, a noticeable disparity in the UMR's trajectory emerges. The presence of the renal circulation leads to a reduction in blood flow past the bifurcation, and as the UMR advances beyond this location, it encounters greater arterial flow. Once the UMR reaches the renal bifurcation, its previously smooth trajectory transforms into a zigzag curve, resulting in increased lateral displacement and, on average, a decrease in swimming speed (Fig. 5A). Alternatively, when the UMR is allowed to swim with the flow (as indicated by the blue trajectories), its propulsive thrust aligns with the direction of the flow, resulting in more seamless swimming behavior, as shown in Fig. 5B. In Fig. 5C, the graph displays the measured distance between the UMR and the centerline of the aorta during this straight run. The UMR exhibits greater lateral displacement when swimming against the flow, especially when it swims closer to the centerline of the

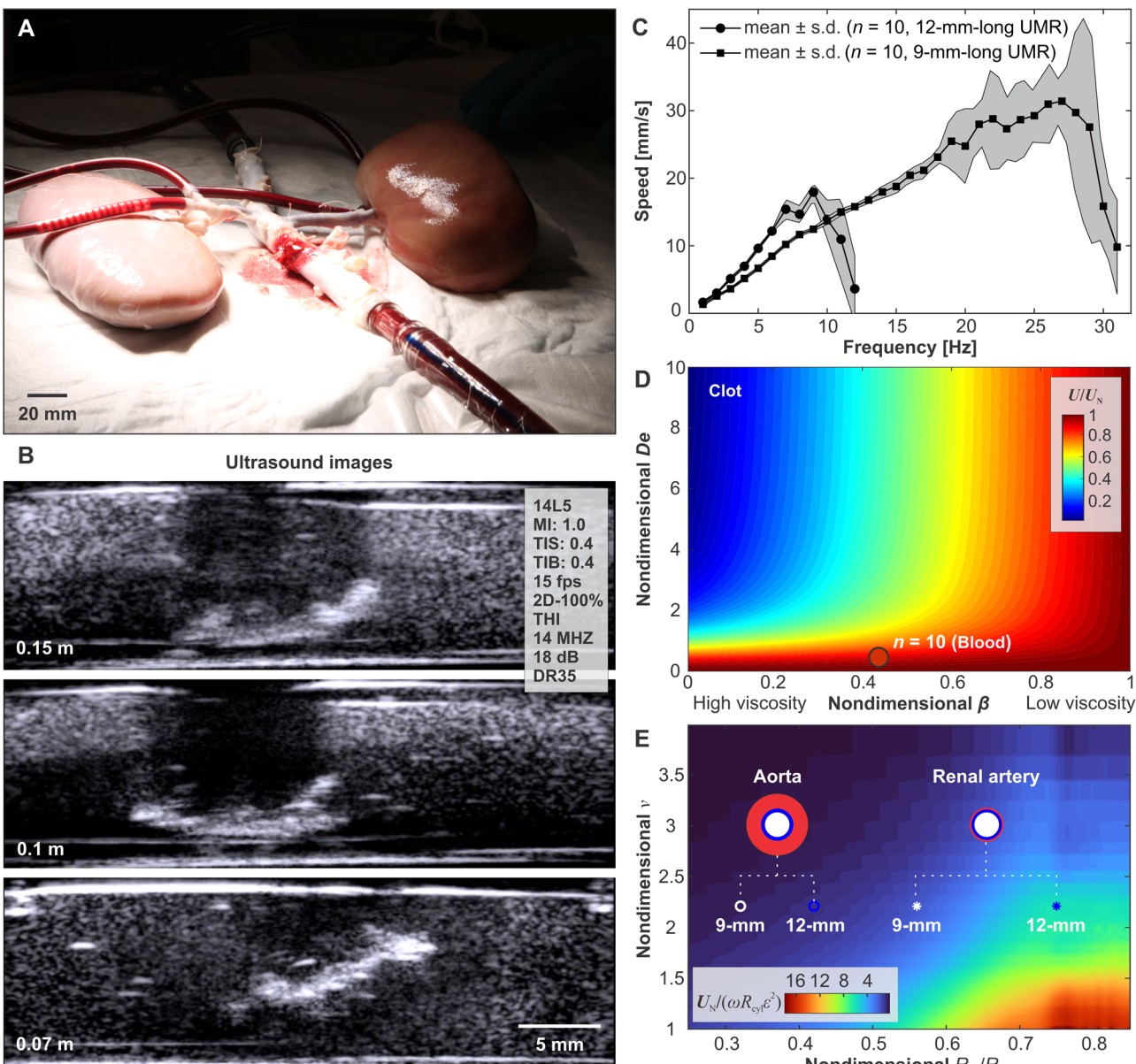

**Fig. 4 | The movement of the untethered magnetic robot (UMR) is influenced by several factors, including the constraints imposed by the ex vivo model's confinement, the viscosity of the blood, and the external control inputs applied.**
**A**, **B** The actuation of the UMR is evaluated by utilizing ultrasound images to identify an optimal gap between the rotating permanent magnet (RPM) and the UMR. This gap is determined to achieve enough RPM clearance while minimizing contact with the inner wall of the lumen. **C** Frequency response of the UMR is characterized in blood. The shaded region represents the standard deviation. **D** Prediction of swimming speed of a UMR in blood and through a clot. The speed of the UMR, $U$, is normalized with swimming speed, $U_N$, in a Newtonian fluid (water). $De$ and $\beta$ are the Deborah number and ratio of serum to blood viscosity, respectively. **E** The UMR's swimming speed is influenced by the normalized wavenumber $v = v^* R_{cyl}$ and diameter of the surrounding vessel ($2R_{ves}$). The white and blue markers indicate the small and large UMRs used in our study, which share the same normalized helical pitch ($2\pi/v$) but differ in their ratios of $R_{cyl}/R_{ves}$.

aorta, where the flow velocity is higher. On the other hand, when the motion is reversed, and the UMR swims with the flow, it becomes more oriented toward the centerline due to the velocity gradient within the aorta.

We gradually increase the pulsatile blood flow and evaluate the straight-run performance of the UMRs both against and with the arterial flow, as shown in Fig. 6 and movie S2. The measured swimming speed of the UMR against the flow exhibits a seemingly linear trend, with the speed decreasing as the flow increases. At a flow rate of 67 mL/hr, the propulsive thrust proves adequate to counteract the flow, although resulting in a small net displacement. In contrast, swimming with the flow results in a speed increase, yet still demonstrates a qualitative correlation with blood flow. Although the average flow rate in the abdominal aorta is 2.9 L/min[29], incorporating additional magnetic material can enhance the UMR's

magnetic moment and step-out frequency. Alternatively, increasing the strength of the external magnetic field can improve propulsive thrust, especially when combined with field-gradient pulling. It's worth noting that UMRs deployed to target clogged vessels (Fig. 1A) are unlikely to encounter such high flow rates since the flow is obstructed.

A plug flow model aligns well with flow rates exceeding 15 mL/min, where the calculated speed, $U$, falls within the range of 16−19 mm/s, under a 9 Hz actuation frequency of the UMR. This is in agreement with the frequency response depicted in Fig. 4C. Additionally, a friction factor ranging from 0.8−1.4 for small UMR and 0.4-0.74 for large UMR estimates a reduction in the UMR's speed attributed to friction with the vessel wall (aorta). In the renal artery, the friction coefficient of the small UMR increased to 1.6, indicating higher friction due to a narrower vessel. The

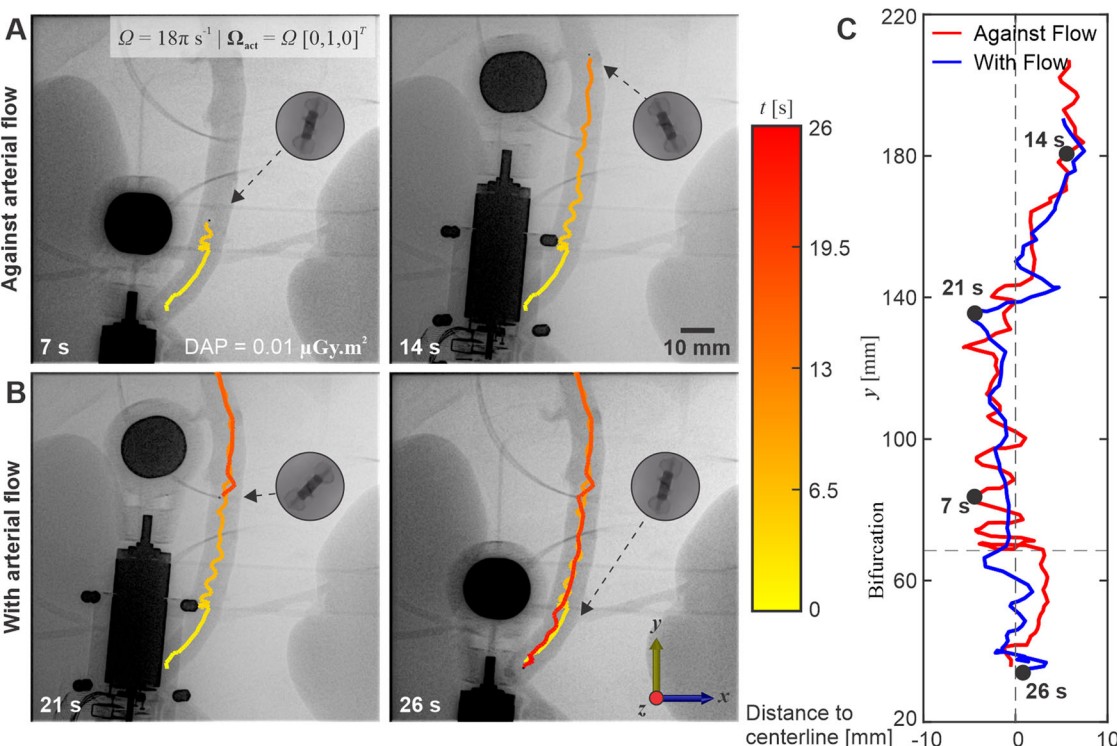

**Fig. 5 | The untethered magnetic robot (UMR) is directed in a controlled manner both against and with the direction of arterial flow, maintaining movement below its step-out frequency.** The UMR is actuated against (**A**) and with (**B**) arterial flow at actuation frequency of 9 Hz. **C** The UMR exhibits greater lateral displacement when swimming against the flow (Movie S2).

swimming speed with flow displays a trend that seems to be less responsive to variations in blood flow. With the UMR and the RPM magnetically interconnected, the field gradient will likely induce a pulling force in the opposite direction of the flow.

**Directional control and steering maneuver into the renal artery**
Guided by the motion characteristics and theoretical model of the UMR, we control the UMR toward the left renal artery to showcase its navigational capabilities. Controlling the magnetic fields is achieved by employing a position-controlled RPM actuator configured in a manner that enables direct teleoperation of its rotation axis, $\Omega_{act}$, and position. Visual tracking of the UMR is facilitated through X-ray Fluoroscopy images captured at a fluoroscopy dose rate of 0.62 mGy cm² s⁻¹ (refer to Methods). Radiation exposure is monitored throughout each trial, ensuring a reduction in the overall radiation dosage, leading to relatively low contrast resolution.

The UMR's passage into the left renal artery along renal flow is achieved through several stages: an initial swimming maneuver toward the renal artery's bifurcation (aligned along the ±$y$ axis within the reference frame of Fig. 7A), a subsequent turning maneuver toward the entry point of the renal artery, followed by a rolling maneuver along the ±$y$ axis, and ultimately a swimming maneuver with the renal flow along the ±$x$ axis (Movies S1 and S3). This four-stage sequence is adopted due to the near right angle formed by the renal artery and the abdominal aorta, making a direct turn substantially challenging. Additionally, due to the inherent limitations of 2D X-ray Fluoroscopy image acquisition, accurately determining the UMR's depth within the abdominal aorta is unfeasible. This, in turn, hinders the ability to make precise adjustments to its height in relation to the entry point of the renal artery using out-of-plane pitch angle swimming.

Screw-shaped UMRs are adept at maneuvering and rolling in proximity to any wall. By rotating the UMR by 90° about the $z$ axis, its long axis becomes perpendicular to the abdominal aorta and aligned with the left renal artery. Consequently, alternating rolling motions around the entry point for roughly 30 seconds results in the successful entry. This rolling is succeeded by swimming and sequential motion reversals within the renal

artery. In this trial, teleoperation guides the UMR to the distal end of the left renal artery in less than 180 seconds. Furthermore, upon internal wall inspection, we observed no indications of damage (as depicted in Fig. 7B).

Figure 7C depicts the sequence of teleoperated inputs that guide the UMR from the abdominal artery to the distal end of the left renal artery. Between $0 < t < 40$ seconds, the UMR initiates swimming against the blood flow along the +$y$ axis, moving past the renal bifurcation. Around $t \sim 30$ seconds, the UMR's direction is reversed, and it is directed to swim towards the renal bifurcation along the −$y$ axis, following the flow. Upon passing the renal bifurcation, the swimming direction is reversed again and the RPM is gradually turned about the $z$ axis to exert an in-plane torque, steering the UMR parallel to the renal artery. Multiple overlaps between the RPM and the UMR obstruct visual feedback during this turning maneuver. This is clearly indicated by the small circles in Fig. 7C between $40 < t < 65$ seconds. Subsequently, the UMR is controlled to roll back and forth between $65 < t < 110$ seconds to enter the left renal artery, assisted by the renal flow.

Once inside the renal artery (around $t \sim 115$ seconds), the UMR swims toward its distal end at a faster speed than that in the abdominal artery. Similarly, when the swimming direction inside the renal artery is reversed (around $t \sim 125$ seconds), and the UMR swims back toward the entry point, its speed is exceeded by that against the arterial flow. This motion enhancement is attributed to the wall effect. With the smaller diameter inside the renal artery, the flow provides an extra force on the body, which increases the swimming speed along flow (Fig. S3).

Figure 8A presents a cross-section view of the renal bifurcation obtained through a CBCT-scan. Clearly, the centerline of the abdominal aorta does not align with the horizontal planes of either the left or the right renal arteries. The left renal artery exhibits a 24° inclination with respect to the horizontal $xy$ plane. Nevertheless, it remains feasible to guide the UMR toward the entry point of the left renal artery and subsequently return to the abdominal aorta. Figure 8B and movie S4 show three consecutive motion control trials directed toward the left renal artery under conditions of stationary blood flow. Throughout these trials, the operator effectively maintained synchronization between the UMR and the RPM, even when faced

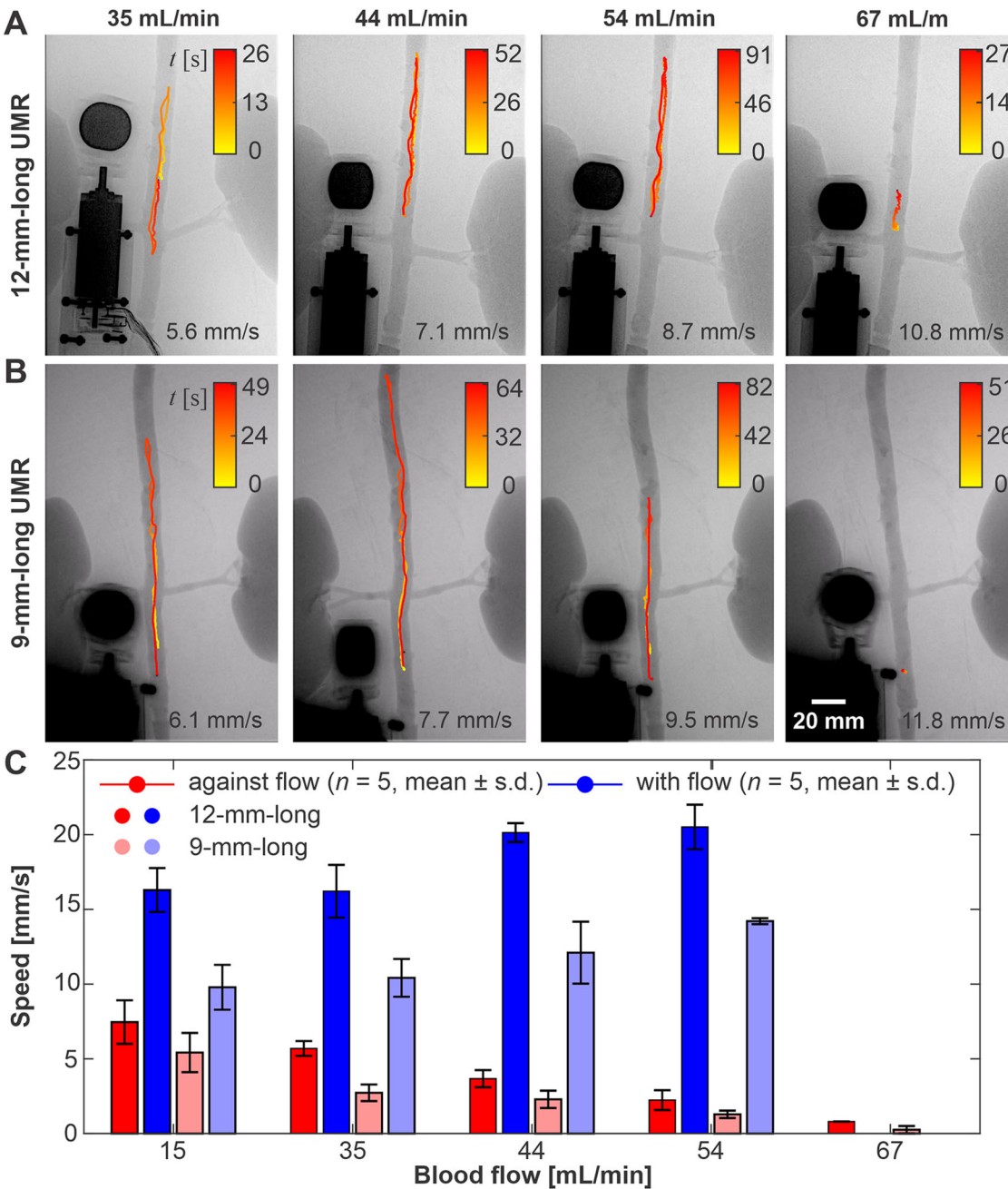

**Fig. 6 | The swimming velocity of the untethered magnetic robots (UMRs) is assessed within the range of blood flow rates from 15 to 67 mL/min. A, B** To achieve consecutive straight runs at an actuation frequency of 9 Hz for each flow rate, the UMR is moved under controlled maneuver. The average speeds are determined based on data collected from five separate trials. **C** The robotic platform effectively maintains the UMR's position against the highest blood flow rate of 67 mL/min (Movie S2).

with temporary visual obstructions (Fig. 8B(i)). To mitigate such visual obstructions, adjusting the UMR's trajectory to enable a direct entry into the left renal artery, as demonstrated in Fig. 8B(ii) and B(iii), can prove to be a highly effective strategy. Direct turns like these are achievable under stationary fluid conditions or with very low flow rates. However, as the flow rate increases, executing a direct 90° turn becomes challenging. In such cases, we employ a combination of rolling and swimming to access the renal artery in less than 60 seconds, as demonstrated in three representative trials in Fig. 8C and movie S4.

## Discussion
In this study, we demonstrate the feasibility of biocompatible UMRs, which are actuated by X-ray-guided magnetic fields. We conduct an in-depth analysis of the UMRs' response using magnetohydrodynamic models, which serves as a basis for selecting design parameters and control inputs for motion control within an ex vivo porcine aorta model. For UMRs that display a small normalized wavenumber (large normalized helical pitch), the cylinder-to-vessel ratio significantly impacts their swimming speed. Conversely, as the normalized wavenumber increases, the influence of confinement diminishes. Based on these theoretical predictions, several design concepts can be proposed to address navigation challenges within varying blood vessel diameters. The first design involves screw-shaped bodies with relatively small normalized helical pitch (large wavenumbers), resulting in slower locomotion but reduced sensitivity to the diameter of the confinement. A second design features screw-shaped bodies with relatively high

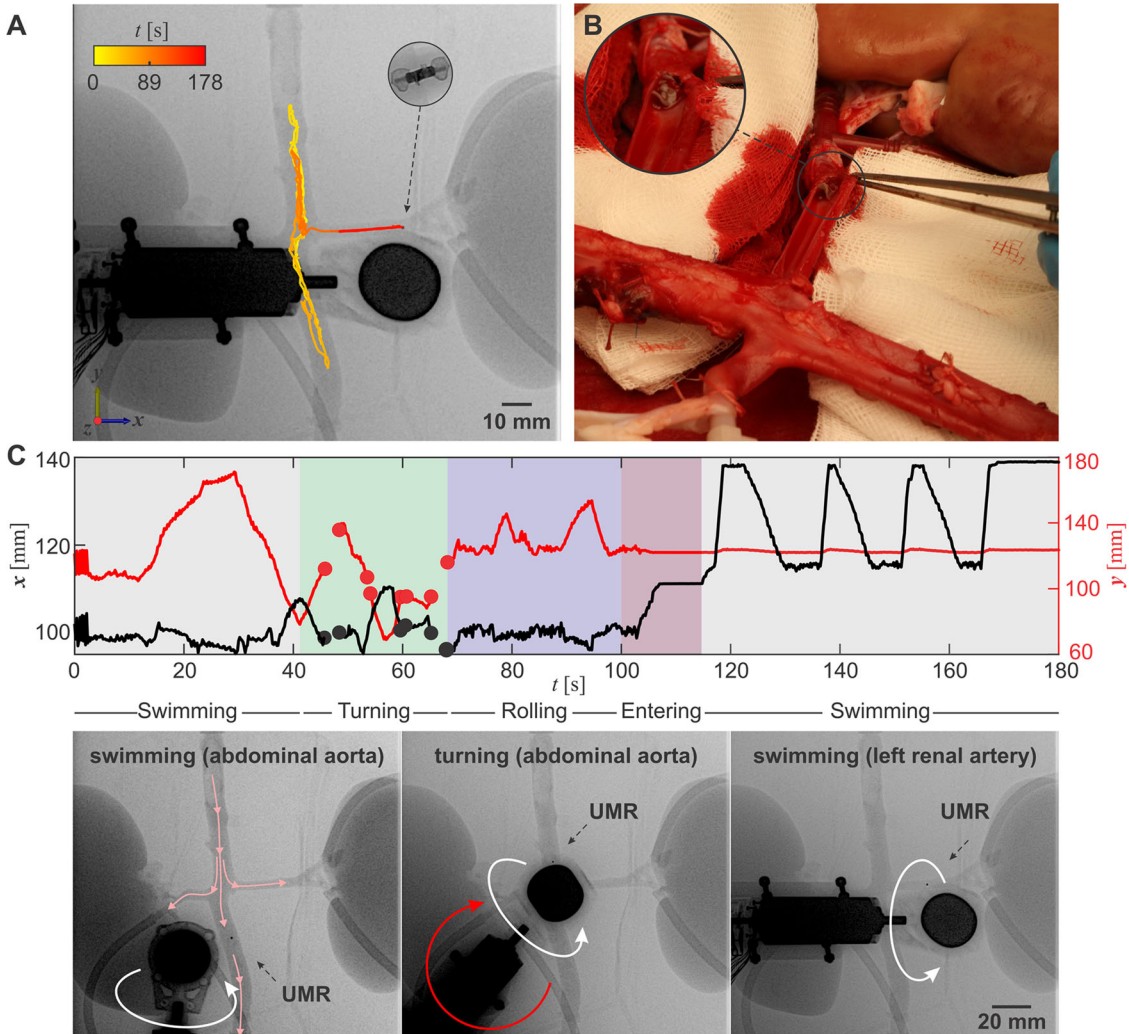

**Fig. 7 | The untethered magnetic robot (UMR) is remotely operated to navigate through the abdominal aorta, engaging in a series of straight runs before executing a turning maneuver within the left renal artery. A** The rotating permanent magnet (RPM) actuator is teleoperated to exert in-plane torque required to steer the UMR toward the left renal artery. **B** The UMR is extracted from the left renal artery and no damage in the wall lining is observed. **C** A four-stage sequence is executed to transition the UMR from its location in the abdominal aorta to the renal artery. The small circles on the visual representation denote instances where the visual feedback of the UMR is obscured by the RPM actuator. The black-dashed arrows indicate the position of the UMR, and the pink arrows indicate the direction of blood flow. The white and red circular arrows indicate the direction of rotation of the RPM with respect to its rotation axis and the $z$ axis, respectively.

normalized helical pitches (small wavenumber), which enhances swimming speed as the cylinder-to-vessel ratio increases, making them well-suited for narrower vessels. Our design blends the robustness of the first type in response to varying blood vessel diameters with the propulsion enhancement anticipated with the low helical pitch and the increasing cylinder-to-vessel ratio of the second type. With this level of control and based on our theoretical predictions, we demonstrate successful direct teleoperation of 9-mm-long UMRs within the abdominal aorta, advancing toward the distal end of the renal artery.

The swimming speeds of our UMRs demonstrate remarkable efficiency, comparable to tethered catheters. It is crucial to emphasize that in medical procedures, clinicians typically have control over the speed at which a catheter is advanced or withdrawn, allowing for adjustments as needed for the specific procedure. When scaled by their body length, our 12-mm-long and 9-mm-long UMRs achieve maximum swimming speeds of 1.6 and 3.3 body lengths per second below their step-out frequencies, respectively. The enhanced efficiency of the smaller UMR is mainly enabled by its reduced resistance to rotation, a characteristic that scales as $R_{cyl}^2$, resulting in a swimming speed that increases linearly with rotational speed $\omega$ and $R_{cyl}$. This scaling effect highlights the advantages of smaller UMRs in achieving

higher relative speeds, comparable to those achieved by tethered devices when controlled by clinicians. However, it is worth noting that catheters have the advantage of exerting much greater force, making them effective for engaging with, for example, thrombus. Currently, our UMRs efficiently harvest magnetic energy and transduce it entirely into work to reach the desired site. Achieving comparable engagements with thrombus as catheters may require significantly greater force. To enhance the frequency response and propulsive thrust of our UMRs, we can explore increasing their magnetic moment and enhancing the strength of the actuating field, which could further improve their performance in targeted applications.

The objective of our UMRs is to provide a minimally invasive solution for treating conditions where blockage cannot be accessed using catheter-based interventions, such as strokes, acute limb ischemia, and chronic limb-threatening ischemia[39]. These conditions often pose challenges for traditional catheter-based interventions due to size constraints and complexities in navigating anatomical structures. By utilizing UMRs, we aim to overcome these limitations and offer more precise and effective treatment options. UMRs have an advantage over catheter-based interventions as they can access locations inaccessible to catheters, such as blood clots below the knee in cases of acute and chronic limb-threatening ischemia. Regarding clot

**Fig. 8 | The 9-mm-long untethered magnetic robot (UMR) is controllably moved back and forth between the abdominal aorta and the proximal end of the left renal artery. A** A cone-beam computed tomography scan shows the $xz$ plane of the renal bifurcation. **B** Motion control is achieved in a stationary blood flow ($\dot{Q} \sim 0$). **C** Motion control is achieved in blood flow of 35 mL/min.

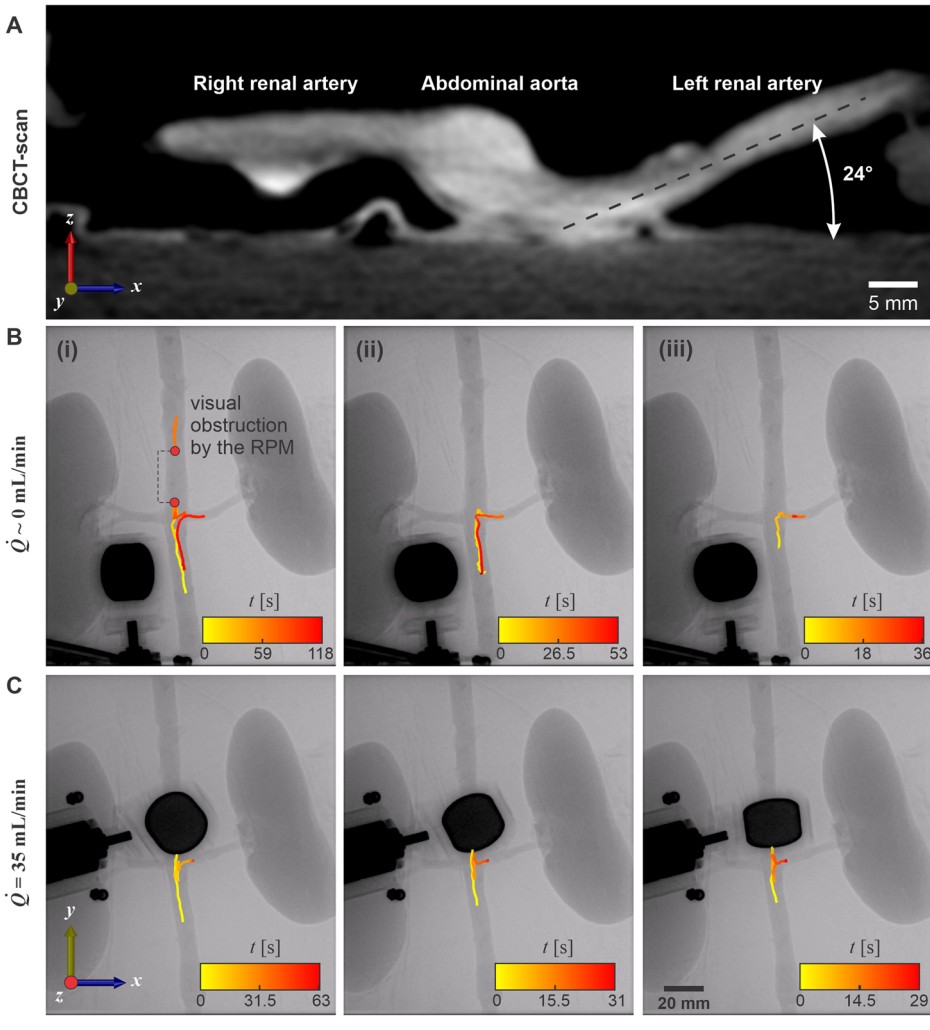

## Materials and methods

### Ex vivo model

Porcine retroperitoneal organs, including the abdominal aorta and kidneys, along with blood, were procured from a slaughterhouse. Following the standard practice for meat processing, the pigs (*Sus scrofa domesticus*, the Netherlands) were rendered unconscious with an electric current applied through electrodes on the body. Subsequently, the carotid artery was incised to drain the animals of blood. The collected blood was directly obtained from the carotid artery wound and placed in a plastic container prepped with 1000 IE heparin per liter of blood (Leo Pharma, Ballerup, Denmark). A total of ten liters of blood was collected from multiple pigs for a single experiment. The retroperitoneal organs, including the aorta and kidneys, along with attached soft tissues, were excised and immediately placed in a plastic bag on ice for transport to the operating room. During transport, the blood was kept at room temperature.

In the operating room, the abdominal aorta, renal vessels, and kidneys were isolated. All aortic side branches, except the renal arteries, were ligated. A standard roller pump perfusion system employing a disposable set of tubing for extracorporeal organ perfusion was utilized. An overflow-secured reservoir was interposed between the pump and the aorta. The system was primed with the heparinized blood. The aorta, with the attached kidneys, was positioned on a plastic sheet and connected to the pump perfusion system by cannulating the proximal and distal ends of the aorta, to allow flow from the proximal end towards the distal end. Both renal arteries were also cannulated and linked to the pump perfusion system to ensure flow towards the kidneys within both renal arteries, while minimizing blood loss through the renal veins. The blood flow was measured at the proximal end to allow experiments at varying flow rates[40].

### In vitro tissue and vessel phantom

We conducted prolonged experiments, including characterizing the frequency response in blood, using an in vitro tissue and vessel phantom. Developing the phantom (275 mm × 85 mm × 40 mm) involved employing a three-piece mold. Initially, the muscle edge layer was positioned on the base plate, and within the muscle edge, vessels (artery: Rehau Silicone 3/8" × 3/32"; vein: Rehau Silicone 3/16" × 1/16") and a femoral nerve (silicone round cord: Ø5 mm), each measuring 350 mm in length, were inserted. To mimic the femoral sheath, an interfacing tissue (80 × 50 mm) was enveloped around the vein and artery, temporarily secured using a paperclip. Following exposure to a heat gun, the paperclip could be detached, allowing the overlapping sides of the tissue to rotate downwards. Mimicking muscle tissue required preparing Ecoflex™ 00-30 with Silc Pig™ Blood pigment, utilizing a WASSERMAN Wamix Touch vacuum mixer. After the curing process, the fat edge layer was added on top of the preceding edge. To mimic fat tissue, Soma Foama™ 15 silicone with Silc Pig™ blood and Silc Pig™ white pigment was prepared using a hand mixer. Following another curing

---

engagement, multiple approaches are feasible. One option is to restore blood flow by drilling through the clot using a screwing action, instantly improving circulation. Alternatively, grinding the clot can rapidly reduce its size. These methods can also be combined with catheter-based thrombolysis to reduce the drug dosage and mitigate negative side effects.

step, the skin edge layer was superimposed onto the previous edge. Subsequently, to replicate skin tissue, Ecoflex™ 00-30 with Silc Pig™ flesh tone was prepared using a WASSERMAN Wamix Touch vacuum mixer. Upon completing the final curing period, the entire mold was extracted.

## Design of the UMRs
The UMRs were designed using three-dimensional computer-aided design software (SolidWorks, Dassault Systèmes, SolidWorks Corp. Inc., USA). The design consists of two identical halves which are joint together while enclosing a permanent magnet. The design finds its origin in previous research where they demonstrated blood clot removal[41]. The length and diameter of the UMRs are based on the inner diameters of the abdominal artery and renal artery, respectively (Table S1), the used parameters can be found in Table S2.

## Fabrication of the UMRs
The UMRs were produced through Masked Stereolithography Apparatus (MSLA) employing a 3D printer (Phrozen Sonic Mini 4K). Phrozen Aqua-Gray 4K resin was the chosen material. During printing, a single layer height of 50 μm was utilized. Subsequent to printing, the components underwent cleaning with isopropyl alcohol (IPA) in an ultrasonic bath for a duration of 7 mins. Post-curing was carried out using an Elegoo Mercury plus curing station for a duration of 12 mins. After curing, UMRs were coated with LipoCoat 4AC coating technology by manual dipcoating for 10 s. Coated UMRs were, then, left to dry overnight under ambient conditions in the dark.

UMRs, possessing a minimum length ($L$) and diameter 9-mm and 3.75 mm, respectively, were constructed by assembling a 3D-printed screw-shaped body with an enclosed permanent magnet composed of NdBFe Grade-N45 material (S-01-01-N Supermagnete, Gottmadingen, Germany). The permanent magnet is cylindrical with axial magnetization, measuring 1 mm in diameter and 1 mm in length, with a magnetic moment ($m$) of $8.4 \times 10^{-4}$ A m². The permanent magnet was positioned such that its magnetic moment was perpendicular to the long axis of the screw-shaped body. This configuration enabled the UMR to swim within blood upon rotation and follow an external weak-strength magnetic field of 5 mT.

## Coating characterization
The successful coating application of LipoCoat 4AC (LipoCoat BV) on UMR material was assessed by comparing coated material to uncoated controls with regard to their water contact angle (CA) using a Krüss DSA30S and to their fluorescence intensity of a fluorescently-labeled coating variant (using an Echo Revolve 4M). Results shown in the SI (Fig. S2) prove the successful coating application. Furthermore, several biocompatibility tests were carried out on coated UMR- and other materials. Protein fouling was tested using a fluorescently labeled fibrinogen variant. Briefly, coated and uncoated UMR material was incubated in PBS containing 0.1 mg/mL fibrinogen-A647 for 2 hr at 37 °C without agitation. After washing in deionized water (MQ), samples were transferred to fresh phosphate-buffered saline for imaging. Average intensities of fibrinogen-A647 adsorbed to coated and uncoated samples were determined using ImageJ[42] and are shown in Fig. 2I.

Biofilm formation was tested on coated, compared to uncoated, PU catheter material using a commercially available *Staphylococcus aureus* strain (ATCC 25923). Briefly, an overnight culture of *S.a.* from Lysogeny broth was washed with M9 minimal salt medium containing 0.4% glucose and 1 mM MgSO₄. Coated and uncoated samples were incubated for 72 h at 37 °C without agitation in the washed o/n culture, diluted with M9 to $OD_{600}$ 0.03. Subsequently, samples were gently washed three times with MQ, fixed by submersion in 4% paraformaldehyde for 10 min, washed with MQ, stained by submersion for 15 min in 3 μg/mL propidium iodide, washed three times with MQ, and imaged dry using a fluorescence microscope. Average intensities of propidium iodide-stained *S.a.* attached to coated and uncoated samples were determined using ImageJ[42] and are shown in Fig. 2J.

A fibrin generation test was carried out in the presence of coated and uncoated UMR material. Briefly, PPP was isolated from freshly drawn, citrated whole blood[36]. Samples were immersed in 65% PPP in 0.9% NaCl, pre-heated to 37 °C, and re-initated by adding 100 mM CaCl₂ (also pre-heated) to reach a final concentration of 14 mM. Immediately after re-initiation, the scatter signal at A₄₀₅ was monitored for 1 h using a Tecan Infinite 200 Pro MPlex. The clotting cascade eventually leads to the formation of fibrin fibres, which causes a sudden increase in light scattering. The scatter signal at 405 nm is monitored in time, starting immediately after initiation. A sharp increase in the scatter signal, denotes the start of fibrin generation and can be accelerated by materials incompatible with the in vivo environment. The resulting time traces and a box plot of their inflection points are shown in Fig. 2K.

Finally, hemocompatibility tests were conducted by HAEMOSCAN BV on the LipoCoat 4AC-coated PU catheter material. These thrombus tests include, visual and gravimetric assessment of thrombi formed on the samples, quantification via immunostaining of fibrin adsorbed to the samples, and enzymatic quantification of attached platelets. Platelet activation was tested by quantifying released thromboxane B2 and beta thromboglobulin as well as platelet aggregation. Coagulation tests consisted of the quantification of thrombin-antithrombin III complex and fibrinopeptide A. Inflammation and complement activation were assessed by quantifying complement component fragments C3a-desArg and C5b-9 as well as elastase.

## Fluidic and structural effects
The UMR is a screw-shaped rigid body with a length of $L$, a diameter of $D$, and an average magnetic moment **m** oriented perpendicular to its long axis. Fluid velocity, vessel walls, and fluid flow within the vessels influence the velocity of the UMR. The body is mathematically represented as a helical wave superimposed onto a cylinder with a radius of $R_{cyl}$, and its surface is described by:

$$\mathbf{x}(\theta, \zeta) = \rho(\theta)[\cos(\nu^*\zeta + \theta)\hat{\mathbf{x}} + \sin(\nu^*\zeta + \theta)]\hat{\mathbf{y}} + \zeta\hat{\mathbf{z}}, \quad (1)$$

where $\theta\epsilon$ [0, 2π] and $\zeta\epsilon$ (−∞, ∞) are helical coordinates and the function $\rho(\theta) = R_{cyl}[1 + \epsilon f(N\theta)]$ describes the profile of the cross-section of the screw-shaped body, and $f(N\theta) = \sin(N\theta)$ is a periodic function, $N$ and $\epsilon$ are the number and the amplitude of starts of the screw. When the UMR submerged in blood is subject to an external magnetic torque, $\mathbf{T} = \mathbf{m} \times \mathbf{B}$, it will move with velocity $U$ and angular rotational rate, $\omega$, satisfying

$$U = 2R_{cyl}\omega\varepsilon^2 \sum_{q \geq 1} \frac{(1 + \beta q^2 De^2)|\hat{f}_q|^2}{1 + q^2 De^2} J_q, \quad (2)$$

where $De = \tau\omega$ is the Deborah number, $\tau$ is the fluid relaxation timescale, and $\beta = \eta_s/\eta$ is a ratio of the blood serum viscosity to the total viscosity of blood[38,43]. The fluid relaxation timescale is estimated from oscillatory shear experiments as

$$\tau = \lim_{\omega \to 0} \frac{G'}{\omega G''}. \quad (3)$$

Here the real and imaginary parts of the complex elastic modulus denoted as $G'$ and $G''$, respectively. In the case of blood, the Deborah number is typically $De = 0.1$[44]. The viscosity of blood serum is ~$1.4-1.5$ mPa s, resulting in a value of $\beta = 0.43$ for blood and $\beta = 0.0015$ for blood clot.

The determination of translational velocity for UMRs within vessels of radius $R_{ves}$ is carried out by distributing Stokeslet points on both the screw-shaped rigid body and the cylindrical vessel surface[45]. Stokeslet points on the screw-shaped rigid body are positioned according to Equation (1) for a total of two turns. These points on the UMR surface possess a velocity of $v(\theta, \zeta) = \omega \times x + U_N$. Meanwhile, the Stokeslet points on the vessel surface remain at a zero velocity. Subsequently, forces acting on the Stokeslet points

are computed, ultimately leading to the determination of the total force acting on the UMR. The translational velocity along the capillary is adjusted until the net force on the UMR reaches zero. This velocity is then determined for various wavenumbers $v^*$ and cylinder-to-vessel ratio $R_{cyl}/R_{ves}$. The utilized UMRs have an $\epsilon$ value of 0.33 and a normalized wavenumber of $v = R_{cyl}v^* = 2.2$. The $R_{cyl}/R_{ves}$ ratio varies with specific UMR applications: The ratio is 0.42 for large UMRs within the aorta, 0.75 for large UMRs in arteries, 0.32 for small UMRs within the aorta, and 0.56 for small UMRs in arteries.

When the UMR moves through a vessel with a flow rate, a plug flow relation is assumed, denoted as $\dot{Q} = \pi R_{ves}^2 U_f$, between the flow rate $Q$ and the fluid flow velocity $U_f$. The UMR's velocity with the flow is given by $U_w = U + U_f - cU_w$, and against the flow, it is $U_a = U - U_f - cU_a$. Here, $U$ represents the UMR velocity in the absence of flow, $cU$ accounts for the friction between the UMR and the vessel wall, and $c$ is the friction coefficient. The friction coefficient $c$ and velocity $U$ can be determined as follows: $c$ is calculated as $c = (2U_f/(U_w - U_a)) - 1$, and $U$ is determined as $U = (c + 1)(U_w + U_a)/2$.

The rheology measurements of our blood indicate viscosities of 27 mPa s and 15 mPa s at room temperature (25 °C) and body temperature (37 °C), respectively, for a shear rate of $2\pi$ s⁻¹. Consequently, the Reynolds number of the 9-mm-long UMR is 0.53 and 0.95 at room and body temperature, respectively, under the influence of an actuating field of 1 Hz. At an actuation frequency of 9 Hz, the Reynolds number increases to 4.2 and 7.6 for room and body temperature, respectively. For the 12-mm-long UMR, the Reynolds number is 0.71 and 1.3 at room and body temperature, respectively, under the influence of an actuation frequency of 1 Hz. With an actuation frequency of 9 Hz, the Reynolds number increases to 8.4 and 15.3 at room and body temperature, respectively.

## Wireless manipulation setup
Wireless actuation is achieved through a robotically controlled RPM actuator. The mechanism generates a rotating magnetic field utilizing a cylinder crafted from NdBFe Grade-N45 material, measuring 35 mm in diameter and 20 mm in height, featuring a magnetic moment of 18.89 A m². The rotational velocity of the permanent magnet is managed using a Maxon 18 V brushless DC motor, while its orientation is regulated via a KUKA 6-DOF manipulator (KUKA KR-10 1100-2, KUKA, Augsburg, Germany). This wireless manipulation carried out within a C-arm fluoroscopic room. The operator is positioned behind a mobile lead barrier for radiation protection, facilitating robotic movement of the RPM.

## Ultrasound imaging
The frequency response of the UMRs in blood was determined by assessing the swimming speed using ultrasound images. For this purpose, the UMRs were placed within the vessel of the in vitro phantom model, characterized by a diameter of 9.5 mm. A 14L5 ultrasound transducer was securely positioned beneath the phantom model, emitting ultrasound waves at a frequency of 11 MHz. A series of consecutive straight runs of the UMR was conducted ($n = 6$) for each actuation frequency of the RPM. The shading displayed in Fig. 4C illustrates the standard deviation (s.d.) in the results.

## Fluoroscopy and cone-beam CT images
The dimensions of the vessels are extracted from a Cone-Beam CT scanner, obtained from three distinct ex vivo animal models. In the teleoperation trials, X-ray Fluoroscopy images are captured utilizing the Siemens Healthineers Artis Pheno (Erlangen, Germany). The image acquisition transpires at a frame rate of 5 Hz, with an X-ray voltage peak of 56.9 kV, tube current of 120 mA. Throughout the trials, no additional contrast media are introduced, barring one in vitro trial (insets in Fig. 3A) where the intention is to depict the UMR's geometry. The wireless actuation employs X-ray Fluoroscopy images deliberately generated with low contrast resolution. This is executed to demonstrate the capability to control the UMR using minimal radiation doses.

## Reporting summary
Further information on research design is available in the Nature Portfolio Reporting Summary linked to this article.

## Data availability
All data needed to evaluate the conclusions in the paper are present in the paper and/or the Supplementary Materials. Additional data related to this paper may be requested from the corresponding author via email.

## Code availability
The code supporting the findings of this study is accessible and available upon request from the corresponding author.

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

## Acknowledgements

This work was supported by the Twente University RadBoudumc Opportunities (TURBO) program 2022, Grant Crazy-Research-2022, and EU-DIRNANO program no. 956544. The authors would like to thank Jaap Greve and Sander Peters for their contributions to the ex vivo experimental results. Figure 1A was created by Now Medical Studios.

## Author contributions

L.-J.W.L. conceived and designed the UMRs and the experiments, characterized the motion characteristics of the UMR, analyzed the data, conducted the motion control experiments, and participated in the writing and editing of the manuscript. N.C.A.R. designed the in vitro experiments, characterized the frequency response of the UMR, and conducted the motion control experiments. C.G. contributed to the rheological characterization of blood. W.C.D., F.R.H., and J.A. designed and developed the in vitro tissue and vessel phantom. R.L. and M.W. conceived and developed the ex vivo porcine aorta model conducted the control experiments and participated in the writing and editing of the manuscript. V.M. participated in the development of the UMRs. A.K. conceived the magneto-hydrodynamic model and participated in the writing and editing of the manuscript. E.A.M.K.R., C.H.E.N., D.W., A.S.-A., and P.J. conceived and developed the coating, conducted the biocompatibility experiments, and participated in writing and editing the manuscript. H.R.L. supervised the X-ray imaging experiments, participated in the motion control experiments, and contributed to writing and editing the manuscript. I.S.M.K. supervised this work, conceived the experiments, analyzed the data, and participated in the writing and editing of the manuscript.

## Competing interests

The authors declare no competing interests.
