## [Peer Review File · Communications Engineering]

Reviewers' comments:

Reviewer #1 (Remarks to the Author):

In this manuscript, the authors study the locomotion and control of a mm-scale helical swimmer in an ex vivo phantom, making use of an aorta, kidneys, and blood harvested from pigs. Ultimately, the authors intend this work to make more plausible the control of similar swimming robots in physiological conditions, including bloodflow, navigating them toward blood clots, using them for mechanical disruption of the clot, and then retrieving them. Among their findings, they note that appropriate surface coating of the swimmers reduces clotting and biofilm formation, that the diameter of the vessel influences swimming speed due to boundary interactions, that they are able to locomote the swimmer against flow, and that in conditions of low flow they can navigate it around sharp corners despite the lack of depth perception offered by x-ray fluoroscopy.

My impression of this work is that it is generally technically sound, reported with sufficient detail to enable reproduction, and brings together many different techniques and apparatuses, such as the permanent magnet RMF generator, the fluoroscope, occasionally ultrasound, etc. For the purpose of publication in Communications Engineering, perhaps this is already sufficient. One critique is that there is relatively little new that is offered here—I am not sure whether anything I read here was particularly surprising or instructive or that the results were striking enough to truly convince me that their swimmer is more ready for deployment in living animals than similar previous work. For one, the swimmer is relatively large, perhaps larger than the kinds of vessels where this kind of clot breaking is needed. The paper is long and contains many figures. While this should not preclude it from publication, I do think that the authors could have considered how to group and more succinctly communicate their most essential results, and e.g. moved many of the fluorographs to the supplementary.

In an effort to help these authors revise their manuscript, I note the following points.

Major comments:

Scale bars should be added throughout the figures, including but not limited to Fig 2i and 2j, Fig 3a, Fig 4b, etc. This is critical for rapidly communicating to the reader what we are seeing.

I feel that comparison to actual physiological values should be made more explicitly and frequently. How does actual flow rate and fluid velocity in the aorta compare to the conditions created here? What is the characteristic diameter of blood vessel clots where this kind of treatment via swimming mili/microrobot would be advantageous over established techniques?

The authors mention “the movement of screw-shaped UMRs within low-Re (on the order of 10–1) blood flow...” However, the swimmer here seems relatively large. Perhaps also calculating or stating the Re here would be beneficial to help connect this particular case with the information that follows.

Minor comments:

The authors state on p5-6 that “for ferromagnetic torque-driven UMRs actuated by homogeneous magnetic fields, MRI systems become unfeasible as high magnetic field associated with imaging would interfere with the magnetization and magnetic response” This is probably an accurate statement, but the authors may also want to consider the dual use of MPI setups for locomotion and sensing. Here is just one example: <https://doi.org/10.1038/s41598-021-93323-4> (There might be others because this is an active area of research in the MPI community.) The x-ray fluoroscopy used in this manuscript obviously provides much higher resolution.

p 3 “this type of experiments” typo, should be “experiment” or “these types”

p 15 “the process [of steering] requires the generation of at least two magnetic torque components simultaneously.” To me, it seems like what they did was change the plane of rotation of their RMF to control direction. In other words, the restoring torque toward the plane of rotation was always present. Perhaps the authors should critically evaluate the phrasing of these sentences to make sure their methodology is communicated clearly.

Throughout the manuscript, “time-periodic” is a strange turn of phrase. “Periodic” would suffice, or “periodically-varying” or “time-varying”. It is not critical to change because the meaning is clear enough, but the phrasing sounds strange.

I suggest rephrasing the term “credibility gap” throughout because I am not sure whether this work really bridges that gap. Maybe something more along the lines of “building toward clinical feasible techniques” would be a bit more generous and realistic as a description.

Extension thoughts (not requiring revision, but maybe worth considering)

On page 11, the authors note that “controlling the UMR is challenging without orientation information”. Could features such as gold nanoparticle markers or something similar be straightforwardly be added to help supply this information?

When I look at the inset of Fig 3A lower inset (of the UMR), what I am most struck by (after its large size, which is hard to deduce without a scale bar) is the wasted space within the UMR. If the plan is just to use this device to mechanically break up clots and then retrieve it, then it would be highly advantageous to have a larger fraction consist of ferromagnetic material. This would allow for higher torques for swimming and drilling. I suppose one issue was buoyancy, but I wonder whether other design aspects could have been changed to improve this.

Reviewer #2 (Remarks to the Author):

Dear authors,

Congratulations to these nice results on steering a micro robot through the aorta of an ex vivo porcine model. This indeed is a step forward for microrobot actuation through the vasculature. To my knowledge the furthest step so far towards in vivo applications. I have some suggestions for making the discussion part stronger:

- Pulsatile flow -> how realistic is the applied flow compared to real flow scenarios

- In this ex vivo scenario, you can bring the magnet much closer to the robot i.e. applying larger forces -> can you comment on what would be the distance in vivo and would it be necessary to use another magnet then?

- To make the article stronger, for me a clear vision is missing what is the purpose of the robot in future? Is it a pathology that you have a thrombus at this site of the body? And this you want to drill through by a robot? Compared to a catheter-based intervention, which problem are you solving with a microrobot?

Please find apart from that some minor comments and suggestions within the attached manuscript.

I recommend this article for publication in Nature Communications Engineering after revision.

Reviewer 1

Comment: In this manuscript, the authors study the locomotion and control of a mm-scale helical swimmer in an ex vivo phantom, making use of an aorta, kidneys, and blood harvested from pigs. Ultimately, the authors intend this work to make more plausible the control of similar swimming robots in physiological conditions, including bloodflow, navigating them toward blood clots, using them for mechanical disruption of the clot, and then retrieving them. Among their findings, they note that appropriate surface coating of the swimmers reduces clotting and biofilm formation, that the diameter of the vessel influences swimming speed due to boundary interactions, that they are able to locomote the swimmer against flow, and that in conditions of low flow they can navigate it around sharp corners despite the lack of depth perception offered by x-ray fluoroscopy.

My impression of this work is that it is generally technically sound, reported with sufficient detail to enable reproduction, and brings together many different techniques and apparatuses, such as the permanent magnet RMF generator, the fluoroscope, occasionally ultrasound, etc. For the purpose of publication in *Communications Engineering*, perhaps this is already sufficient. One critique is that there is relatively little new that is offered here—I am not sure whether anything I read here was particularly surprising or instructive or that the results were striking enough to truly convince me that their swimmer is more ready for deployment in living animals than similar previous work. For one, the swimmer is relatively large, perhaps larger than the kinds of vessels where this kind of clot breaking is needed. The paper is long and contains many figures. While this should not preclude it from publication, I do think that the authors could have considered how to group and more succinctly communicate their most essential results, and e.g. moved many of the fluorographs to the supplementary.

In an effort to help these authors revise their manuscript, I note the following points.

Response: Thank you for your feedback. We greatly appreciate the time and effort you have dedicated to reviewing our manuscript. We have incorporated all your suggestions into the revised version of the manuscript.

We have reorganized certain descriptions related to hemocompatibility and fluidic properties into the Materials and Methods section, which now appear on *Page 30*, *Page 31*, and *Page 33* of the revised manuscript. Our manuscript comprises 8 Figures strategically arranged to highlight the core findings of the study, including the development of an X-ray-guided robotic platform and *ex vivo* system, wireless locomotion of UMRs in arterial flow, and directional control and steering of UMRs toward hard-to-reach locations within the ex vivo model.

Figure 1 is pivotal as it illustrates the vision and potential of the UMR to access inaccessible locations, supporting the problem formulation through the depiction of the intended path in the CBCT scans. Figures 2, 3, and 4 address the development of the ex vivo model and its integration within an X-ray-guided robotic platform. They also provide essential details regarding the detectability of the UMR, RPM-actuator, and the anatomical information of the environment. Additionally, the results in Figure 4 offer important theoretical and experimental predictions about the impact of the rheological properties of the fluid and the proximity to a wall on the swimming speeds of the UMR. Figures 5 and 6 show the response of the UMR against and along arterial flow, highlighting limitations that need to be considered during motion control. Finally, Figures 7 and 8 demonstrate how

telemanipulation is performed, enabling the UMR to enter the renal artery and reach its distal end under different flow conditions.

Comment: Scale bars should be added throughout the figures, including but not limited to Fig 2i and 2j, Fig 3a, Fig 4b, etc. This is critical for rapidly communicating to the reader what we are seeing.

Response: In the revised manuscript, we have included scale bars across all figures.

Comment: I feel that comparison to actual physiological values should be made more explicitly and frequently. How does actual flow rate and fluid velocity in the aorta compare to the conditions created here? What is the characteristic diameter of blood vessel clots where this kind of treatment via swimming mili/microrobot would be advantageous over established techniques?

Response: The realistic average flow rate inside the abdominal aorta is approximately 2.9 L/min [1]. Our experiments demonstrate that at an actuation frequency of 9 Hz, both the 9-mm-long and 12-mm-long UMRs are capable of overcoming blood flow rates of up to 67 mL/min, as depicted in Figure 6C of the revised manuscript. Based on the frequency response of the 9-mm-UMR (Figure 4C), we anticipate that the UMR could overcome flow rates of up to 180 mL/min at an actuation frequency of 28 Hz (Figure 4C in the revised manuscript). To further enhance their performance, we can increase the magnetic moment by augmenting the fraction of ferromagnetic material within the UMRs and reduce their size. For instance, by adding an extra permanent magnet, as demonstrated in Fig. 1 of this response letter, the UMR can navigate against flow rates of up to 250 mL/min at an actuation frequency of 40 Hz. Additionally, we can utilize magnetic field gradient pulling to stabilize the UMR against stronger blood flow, thereby improving its navigation efficiency. Navigating the UMR within the abdominal aorta presents significant engineering challenges due to the high flow rates encountered. We acknowledge the ongoing need for optimization to continually enhance the capabilities of our system in the future. Nevertheless, it is expected that the blood flow rate will significantly decrease in the presence of a blockage, and therefore the UMR will encounter less flow rate in these scenarios where it is intended to target a clogged vessel.

Fig. 1. Incorporating two permanent magnets within the UMR enhances its swimming speed and enables it to overcome flow rates of up to 250 ml/min at actuation frequency of 40 Hz. This new data set has been incorporated into the revised *Supplementary Materials* as *Figure S1*.

Renal artery thrombosis, acute limb ischemia (ALI), and chronic limb-threatening ischemia (CLTI) often involve the formation of blood clots that are challenging to access using tethered flexible systems. These clots typically range in diameter from 4 mm to 10 mm, making them difficult to reach with traditional methods. However, our magnetically-guided UMR can be precisely steered toward these clots to restore blood flow effectively. The renal artery typically has a diameter in the range of 4-7 mm, while peripheral arteries affected by ALI and CLTI have an average diameter of 10 mm. In such cases, the maneuverability and precision offered by our UMR provide a distinct advantage in reaching and addressing these clot-related issues to restore vital blood flow. We have included these explanations on *Page 9* and *Page 19* of the revised manuscript, and we have also provided additional supplementary material (Fig. S1) to demonstrate the enhanced frequency response achieved when increasing the magnetic material within the UMR.

[1] Taylor, C.A., Cheng, C.P., Espinosa, L.A. et al. In Vivo Quantification of Blood Flow and Wall Shear Stress in the Human Abdominal Aorta During Lower Limb Exercise. *Annals of Biomedical Engineering* 30, 402–408 (2002).

The authors mention “the movement of screw-shaped UMRs within low-Re (on the order of 10–1) blood flow...” However, the swimmer here seems relatively large. Perhaps also calculating or stating the Re here would be beneficial to help connect this particular case with the information that follows. **Response:** We appreciate the valuable suggestion from the reviewer. Despite our UMRs being in the millimeter scale, they are actuated using low-frequency magnetic fields, as illustrated in Figure 4C in the revised manuscript. The rheology measurements of our blood indicate viscosities of 27 mPa.s and 15 mPa.s at room (25°C) and body (37°C) temperature, respectively, for a shear rate of $2\pi/s$. Consequently, the Reynolds number of the 9-mm-long UMR is 0.53 and 0.95 at room and body temperature, respectively, under the influence of an actuating field of 1 Hz. At an actuation frequency of 9 Hz, the Reynolds number increases to 4.2 and 7.6 for room and body temperature, respectively. For the 12-mm-long UMR, the Reynolds number is 0.71 and 1.3 at room and body temperature, respectively, under the influence of an actuation frequency of 1 Hz. With an actuation frequency of 9 Hz, the Reynolds number increases to 8.48 and 15.3 at room and body temperature, respectively. We have therefore indicated that the Reynolds number changes in the range between 0.1 to 20 in the revised manuscript on *Page 14* and *Page 33*.

Minor comments:

Comment: The authors state on p5-6 that “for ferromagnetic torque-driven UMRs actuated by homogeneous magnetic fields, MRI systems become unfeasible as high magnetic field associated with imaging would interfere with the magnetization and magnetic response” This is probably an accurate statement, but the authors may also want to consider the dual use of MPI setups for locomotion and sensing. Here is just one example: <https://doi.org/10.1038/s41598-021-93323-4> (There might be others because this is an active area of research in the MPI community.) The x-ray fluoroscopy used in this manuscript obviously provides much higher resolution.

Response: Thank you for your valuable input and for providing the reference to the Magnetic Particle Imaging (MPI) setup for locomotion and sensing (Reference [24] in the original and revised manuscripts). While MRI systems may face challenges due to interference with ferromagnetic torque-

driven UMRs, we acknowledge the potential of MPI setups for dual use in locomotion and sensing, as demonstrated in the referenced study. However, it is important to note that MPI is a tomographic method specifically designed to detect the three-dimensional distribution of superparamagnetic iron-oxide nanoparticles (SPIONs). Unlike other imaging modalities, MPI signal is solely derived from the particles themselves and not from surrounding tissue. Therefore, our focus on X-ray Fluoroscopy imaging in this manuscript is primarily driven by its ability to provide anatomical information at higher resolution, which better suits our specific experimental needs. Integrating rotating magnetic field control with an additional tomographic modality enables the registration of anatomical information, thereby enhancing the diagnostic potential and motion control capabilities because the surrounding tissue, the target, the UMR-actuator, and the UMR are captured in each CT scan. We have provided these details in the revised manuscript on *Page 6* and *Page 10*.

Comment: p 3 “this type of experiments” typo, should be “experiment” or “these types”

Response: We have corrected this typographical error on *Page 3* in the revised manuscript.

Comment: p 15 “the process [of steering] requires the generation of at least two magnetic torque components simultaneously.” To me, it seems like what they did was change the plane of rotation of their RMF to control direction. In other words, the restoring torque toward the plane of rotation was always present. Perhaps the authors should critically evaluate the phrasing of these sentences to make sure their methodology is communicated clearly.

Response: We agree with the reviewer's observation that the UMR can achieve either a pitching motion (relative to the x-axis in the frame of reference in Figure 3D) or a steering motion (relative to the z-axis) by controlling the plane of rotation of the magnetic field, or by controlling the rotating permanent magnet (RPM)'s rotation axis, as depicted in Fig. 3C and 3D. We have provided these explanations on *Page 14* in the revised manuscript.

Comment: Throughout the manuscript, “time-periodic” is a strange turn of phrase. “Periodic” would suffice, or “periodically-varying” or “time-varying”. It is not critical to change because the meaning is clear enough, but the phrasing sounds strange.

Response: We have used "periodic" throughout the revised manuscript.

Comment: I suggest rephrasing the term “credibility gap” throughout because I am not sure whether this work really bridges that gap. Maybe something more along the lines of “building toward clinical feasible techniques” would be a bit more generous and realistic as a description.

Response: Thank you for the valuable suggestion. We have updated the sentence as per your recommendation in the revised manuscript on *Page 5*.

Extension thoughts (not requiring revision, but maybe worth considering)

Comment: On page 11, the authors note that “controlling the UMR is challenging without orientation information”. Could features such as gold nanoparticle markers or something similar be straightforwardly be added to help supply this information?

Response: We appreciate the reviewer's insightful suggestion. Incorporating radiopaque objects, such as gold nanoparticle markers, into the UMR can indeed provide valuable orientation information. We have provided this explanation and suggestion on *Page 12*, in the revised manuscript.

Comment: When I look at the inset of Fig 3A lower inset (of the UMR), what I am most struck by (after its large size, which is hard to deduce without a scale bar) is the wasted space within the UMR. If the plan is just to use this device to mechanically break up clots and then retrieve it, then it would be highly advantageous to have a larger fraction consist of ferromagnetic material. This would allow for higher torques for swimming and drilling. I suppose one issue was buoyancy, but I wonder whether other design aspects could have been changed to improve this.

Response: Thank you for highlighting the complexities related to the size and design of the UMR. While increasing the fraction of ferromagnetic material could enhance its ability to navigate through stronger blood flow, as shown in Figure 1 in this letter (Fig. S1 in the supplementary materials), we must also address the buoyancy issue. This is particularly critical during transitions between larger and smaller blood vessels, such as the entrance of the renal arteries from the abdominal aorta, as depicted in Figure 2 (in this response letter) and Figure 8A (in the revised manuscript). When the weight of the robot is increased, it tends to roll on the lumen rather than swim, impacting its steering capabilities. This buoyancy effect poses challenges for swimming and steering, necessitating careful consideration in our design optimizations.

Fig. 2. Increasing the weight of the UMR results in continuous contact with the lining of the lumen. Consequently, steering the robot with respect to the z-axis to enter either the left or the right renal arteries is adversely affected because contact with the lumen induces undesirable rolling locomotion.

Reviewer 2

Comment: Congratulations to this nice results on steering a micro robot through the aorta of an ex vivo porcine model. This indeed is a step forward for microrobot actuation through the vasculature. To my knowledge the furthest step so far towards in vivo applications. I have some suggestions for making the discussion part stronger:

- Pulsatile flow -> how realistic is the applied flow compared to real flow scenarios

Response: The average flow rate within the abdominal aorta is approximately 2.9 L/min [1]. At an actuation frequency of 9Hz, both the 9-mm-long and 12-mm-long UMRs can navigate against a blood flow of 67 mL/min (as depicted in Figure 6C in the revised manuscript). According to the frequency response of the 9-mm-UMR (illustrated in Figure 4C), it is anticipated that the UMR could navigate against a flow rate of 180 mL/min at an actuation frequency of 28 Hz. Enhancing the magnetic moment (by increasing the fraction of ferromagnetic material) and reducing the size of the UMRs can enhance their frequency response, allowing them to navigate against higher flow rates. For instance, we've integrated an extra permanent magnet into a UMR and measured its swimming speed, as shown in Fig. 1 in this response letter. Consequently, at an actuation frequency of 40 Hz, the UMR can navigate against a flow of 250 mL/min. It is worth noting that magnetic field gradient pulling can be employed to hold the UMR against stronger blood flow, enabling it to navigate through realistic flow rates. The control of UMR motion within the abdominal aorta poses a significant engineering challenge due to the substantial flow rates. Our study represents an initial step in addressing these challenges, and we recognize the need to optimize our system for improved performance in the future. Nevertheless, it is not expected that the UMR will encounter such high arterial flow rates when blood clots and vessels are clogged. The clogged vessel will exhibit lower flow rates than those encountered in the abdominal aorta. We have provided these explanations on *Page 9* of the revised manuscript.

[1] Taylor, C.A., Cheng, C.P., Espinosa, L.A. et al. In Vivo Quantification of Blood Flow and Wall Shear Stress in the Human Abdominal Aorta During Lower Limb Exercise. *Annals of Biomedical Engineering* 30, 402–408 (2002).

Comment: - In this ex vivo scenario, you can bring the magnet much closer to the robot i.e. applying larger forces -> can you comment on what would be the distance in vivo and would it be necessary to use another magnet then?

Response: We thank the reviewer for the valuable suggestion. The average *in vivo* gap from the abdominal aorta to the skin is 12 cm. Therefore, we have limited the the RPM-UMR gap to 12 cm in our experiments. It is possible to increase this distance by using an RPM with greater magnetic moment and it is also possible to exploit the magnetic field gradient pulling to assist the propulsive thrust. Currently our UMRs swim using homogenous rotating fields and they are subject to minimal field gradient pulling as they swim in sync with the RPM. We have provided these explanations on *Page 16* in the revised manuscript.

Comment: - To make the article stronger, for me a clear vision is missing what is the purpose of the robot in future? Is it a pathology that you have a thrombus at this site of the body? And this you want to drill through by a robot? Compared to a catheter-based intervention, which problem are you solving with a microrobot?

Response: Thank you for your insightful comment. The purpose of our microrobot is to offer a

minimally invasive solution for treating conditions such as thrombosis, acute limb ischemia, and chronic limb-threatening ischemia, where traditional catheter-based interventions face limitations such as size constraints or difficulties navigating complex anatomical structures. By employing microrobots, we aim to surmount these challenges and provide more precise and effective treatment options for such pathologies. Therefore, we have included Figure 1A to illustrate our vision. This figure depicts a scenario where a UMR can be deployed and maneuvered toward a blockage located within a branched vessel. UMR has an advantage over catheter-based intervention that they can access locations inaccessible to catheters. For example, blood clots developed below the knee in the case of acute limb ischemia and chronic limb-threatening ischemia cannot be reached by catheters.

Fig. 3. Blood clots often form in regions inaccessible to flexible instruments. In such cases, untethered magnetic robots can be deployed to precisely target these locations.

The approach of engaging with the blood clot is not exclusive. It is feasible to restore blood flow by drilling through the clot using a screwing action, instantly improving circulation. Another option is to grind the clot to reduce its size quickly. Alternatively, clot grinding can be combined with catheter-based thrombolysis to reduce the dosage of the drug and mitigate associated negative side effects. We have provided these explanations and discussions into the revised manuscript on *Pages 26 and 27*.

Comment: Please find apart from that some minor comments and suggestions within the attached manuscript. I recommend this article for publication in *Nature Communications Engineering* after revision.

Response: We appreciate the thorough feedback provided by the reviewer in the PDF file. Below, you will find comprehensive responses addressing each comment in detail.

Comment: Page 2 on the abstract, where the authors state: “external wireless forces and torques”; Do you mean already magnetic forces and torques? If so, insert "magnetic" if you want to stay more

general in this sentence, then it comes a bit surprisingly that you are talking about magnetic robots in the next sentence.

Response: In the revised manuscript on *Page 2*, we have updated it to "external magnetic forces and torques," as advised by the reviewer.

Comment: Page 4 in the description of Figure 1, the authors state " The wireless actuation and non-invasive localization of UMRs are achieved through a robotic platform "; please be a bit more precise on that. At this stage in the manuscript readers don't understand how you realized actuation and especially localization by just naming it "robotic platform"

Response: Thank you for highlighting this issue. Since the actuation system has not been discussed yet, we've made the required adjustment by adding ", consisting of an external magnetic field and a C-arm imaging system," to the caption of Figure 1 on *Page 4*. This addition offers readers valuable insight into our actuation and localization platform.

Comment: On page 6 where the authors discuss different localization methods, the reviewer mentions: What about other imaging techniques? You don't need to be such detailed as before, but aren't there also approaches using CT, x-ray, PET, MPI, Photoacoustic,...? Especially since you are using CT, would be interesting if this technique has been used before or what is different in your work.

Response: We are grateful to the reviewer for their valuable suggestions. In response, we have revised the description of the various localization methods utilized in localizing and controlling untethered devices, as outlined in the revised manuscript on *Page 6*. Furthermore, we have provided detailed explanations regarding the utilization of magnetic particle imaging (MPI) scanners to facilitate wireless actuation of untethered devices.

Comment: On page 6 the authors refer to "Figs 1B and 1C (Movie S1)"; in the movie the UMR is moving several times back and forth, that is not shown in the figure. Either say in Fig 1 the planned path is shown and in the movie the resulting trajectory. Or also show a screenshot of the movie in the figure. I would prefer the second option

Response: We have clarified that the yellow arrow in Figure 1B represents the planned path, not the actual path taken by the UMR. The actual path is depicted in Figure 7A. Furthermore, Figures 1B and 1C display CBCT scans indicating the final position of the UMR, denoted by the black and white squares. These clarifications have been included on *Page 6* of the revised manuscript.

Comment: On page 8 the authors mention " The proximal and distal ends of the abdominal aorta are connected to a peristaltic pump for blood circulation at controlled flow rate in the $15 \leq Q \leq 260$ mL/min range " Could you please clarify if it's a pulsatile flow and what are the flow rates in vivo?

Response: Blood circulation is accomplished through a peristaltic, producing a controlled pulsatile flow. We have provided this explanation in the revised manuscript on *Page 9*.

The average flow rate in the abdominal aorta is 2.9 L/min. However, this flow rate is not likely to be achieved when the blood vessel exhibits a blockage. Additionally, using heart rate-lowering drugs it is possible to decrease the blood flow to be able to control the UMR and it is also possible to increase the propulsive thrust of the UMR by increasing its ferromagnetic material as shown in Fig. 1 in this response letter. We have provided these explanations on *Page 9* in the revised manuscript.

Comment: On page 9 the authors mention "A rotating magnetic field", it is suggested that the authors specify that it is a rotating magnetic field gradient

Response: Thanks to the reviewer for pointing this out, in this case we now have specified that it is a magnetic field gradient on *Page 10*.

Comment: On Figure 3 this arrow connection is misleading since one image is ex vivo and the other inside a tube.

Response: We have deleted this arrow in the revised manuscript on *Page 11*.

Comment: On page 11 the authors mention " Consequently, solely the magnet of the UMR becomes visible in the X-ray Fluoroscopy images in Figs. 3A, 3C, and 3D. "; only the magnet would become (replace solely, since it was used before and the images shown are with contrast agent, so make clear that this is hypothetical, but not what is shown in the figure.

Response: In a clinical setting, only the magnet will produce a detectable signal, as illustrated in Figs. 3A, 3C, and 3D, unless contrast agents are introduced into the blood, as depicted in the inset of Fig. 3. We concur with the reviewer that this aspect might not have been clear to the readers. To address this, we have reordered the sentences in the corresponding paragraph on *Pages 10-12* of the revised manuscript, placing greater emphasis on the detection of only the magnet. Subsequently, we explain how the inset of Figure 3 is generated.

Comment: On the results of the Hemobiocompatibility of UMRs on pages 11-13: in my opinion in these two sections some details could be moved to the methods section

Response: We appreciate the reviewer's valuable suggestion. In response, we have condensed the section on Hemobiocompatibility of UMRs, now located on *Pages 12 and 13*, and relocated the detailed information to the Methods Section on *Page 31* in the revised manuscript.

Comment: On Figure 4D: Could you please state in the caption what is De and β ?

Response: De and β are the Deborah number and the ratio of blood to serum viscosity, respectively. We have revised the caption of Figure 4 in the revised manuscript on *Page 15*.

Comment: On page 15 where the authors mention " Fig. 4B displays ultrasound images of a UMR designed to fit through the renal artery upon being deployed in the abdominal aorta. "; the US

technique jumps a bit surprisingly into the story line... before you just explained and mentioned the CT, could you please state why here US is used and how. I miss here a bit an introductory sentence bridging this gap.

Response: The mentioned figure is meant to illustrate the position of the UMR with respect to the centerline of the vessel at different RPM-UMR gaps. Here it is important that due to the change in magnetic field, the UMR gets attracted away from the bottom of the vessel at low RPM-UMR gap. Since this measurement has to be taken in blood due to the buoyancy and has to be taken in the vertical plane, camera or X-ray would not be a suitable option. Therefore, ultrasound has been utilized to create this figure. To make this clearer to the reader, the order of a few sentences is changed, and ultrasound is now merely mentioned as a tool for visualizing the effect of the RPM-UMR gap, as shown on *Page 16* of the revised manuscript.

Comment: On page 17 the authors mention " R_{vec} ", why " vec " doesn't it stand for vessel

Response: We have corrected this typographical error on *Page 17* and other instances in the revised manuscript.

Comment: On page 18 the authors mention "resulting in increased horizontal displacement and, on average, a decrease in swimming speed (Fig. 5A). "; forward would be also horizontal. In the figure caption you used "lateral". I would prefer keeping the same wording

Response: Thank you for the feedback. We have revised the text on *Page 18* accordingly, replacing "horizontal" with "lateral" to avoid confusion.

Comment: On page 20 the authors mention "A plug flow model aligns well with flow rates exceeding 15 mL/min, where the calculated speed, U , falls within the range of 16 – 19 mm/s, under a 9 Hz actuation frequency of the UMR. These values are depicted in Fig. 4. " Where? I don't know where exactly you are pointing at here.

Response: In this section we discuss the agreement between the model and the experimental values. From the plug flow model followed that the speed of the UMR would be in the range of 16-19 mm/s which agrees with our frequency response at 9 Hz, around the step-out frequency of the 12-mm-long UMR, where we achieved a speed of 17.9 ± 1.2 mm/s. In the revised manuscript we make this clearer by specifying Fig 4C, see *Page 19*.

Comment: On Figure 7: where is the arrow pointing at?

Response: The black-dashed arrows point toward the UMR that is inside the abdominal aorta and swim toward the renal artery. Please note that the body of the UMR is radio-opaque and therefore it is only possible to determine the position of the permanent magnet inside its structure. The arrows point toward the permanent magnet of the UMR. The pink curved arrows indicate the direction of the blood flow inside the abdominal aorta and the renal arteries. We have provided these descriptions in the caption of Figure 7, in the revised manuscript on *Page 22*.

Comment: On page 26 where the materials and methods of the *ex vivo* model are discussed: I am missing here how you connected the *ex vivo* model to tubes and to a pump and how you pumped the blood through the model.

Response: We have added more information of the materials we used, and which ends of the perfusion model were connected to the pump system. Also, the direction of the flow, which is in the physiologically accurate direction, has now been specified. See *Pages 27 and 28* in the revised manuscript.

Comment: Page 29, "ImageJ (?)" is highlighted

Response: We have corrected this reference error on *Page 30* in the revised manuscript.

Comment: Page 31 " f is calculated as $c = (2Uf/(Uw - Ua)) - 1$ ", is highlighted

Response: We have corrected this typographical error on *Page 33* in the revised manuscript.

Comment: On the supplementary Figure S1, Especially for B, but maybe also for D applicable: how many measurements? So what is n in your measurements?

Response: We have added the number of measurements to Figure S1 in the revised Supplementary Materials (Fig. 4 in this response letter).

Fig. 4. Validation of coating presence on UMR material is depicted as follows: (A) Images display sessile drops of deionized water from contact angle measurements on uncoated and coated samples.

REVIEWERS' COMMENTS:

Reviewer #1 (Remarks to the Author):

The authors have incorporated numerous thoughtful revisions based on my and the other reviewer's comments, as well as small updates to figures. In cases where they did not make changes (e.g. the number of figures), well-reasoned justification can be found in the point-by-point response document. In reviewing these documents, it is my view that the revisions would be adequate for publication.

Congrats on the nice manuscript and best of luck to these authors in this work and their future work.

Reviewer #2 (Remarks to the Author):

All concerns have been addressed by the authors and the manuscript has been edited accordingly. I suggest the article for publication.